# Activin promotes skin carcinogenesis by attraction and reprogramming of macrophages

Maria Antsiferova[1], Aleksandra Piwko-Czuchra[1], Michael Cangkrama[1], Mateusz Wietecha[1], Dilara Sahin[1], Katharina Birkner[1], Valerie C Amann[2], Mitchell Levesque[2], Daniel Hohl[3], Reinhard Dummer[2] & Sabine Werner[1,*]

## Abstract

Activin has emerged as an important player in different types of cancer, but the underlying mechanisms are largely unknown. We show here that activin overexpression is an early event in murine and human skin tumorigenesis. This is functionally important, since activin promoted skin tumorigenesis in mice induced by the human papillomavirus 8 oncogenes. This was accompanied by depletion of epidermal γδ T cells and accumulation of regulatory T cells. Most importantly, activin increased the number of skin macrophages via attraction of blood monocytes, which was prevented by depletion of CCR2-positive monocytes. Gene expression profiling of macrophages from pre-tumorigenic skin and bioinformatics analysis demonstrated that activin induces a gene expression pattern in skin macrophages that resembles the phenotype of tumor-associated macrophages in different malignancies, thereby promoting angiogenesis, cell migration and proteolysis. The functional relevance of this finding was demonstrated by antibody-mediated depletion of macrophages, which strongly suppressed activin-induced skin tumor formation. These results demonstrate that activin induces skin carcinogenesis via attraction and reprogramming of macrophages and identify novel activin targets involved in tumor formation.

**Keywords** activin; macrophage; skin cancer; tumor microenvironment
**Subject Categories** Cancer; Immunology; Skin

## Introduction

Epithelial skin cancers, in particular basal and squamous cell carcinomas (BCC and SCC), are the most frequent types of cancer in humans, and they are diagnosed in 2–3 million people worldwide every year. Due to the increased life expectancy, the enhanced sun exposure, and the treatment of organ transplant patients with immunosuppressive compounds, the incidence of epithelial skin cancer is continuously increasing (Griffin et al, 2016). Cutaneous SCCs often develop from precursor lesions, of which actinic keratosis (AK) is the most frequent one. Thus, it has been estimated that 65% of cutaneous SCCs arise from AK (Criscione et al, 2009). AK develops at multiple sites, in particular in sun-exposed skin, and its treatment is important due to the risk of malignant transformation (Werner et al, 2013). Importantly, it is generally difficult to predict whether an AK lesion will progress to SCC, and it is therefore important to identify biomarkers that indicate a high risk for malignant progression. This requires a thorough understanding of the pathomechanisms underlying AK development and progression and the identification and functional characterization of the involved cell types and genes/proteins. Some of them may represent novel targets for therapeutic intervention. They include proteins that directly affect proliferation, survival and migration of keratinocytes and their malignant transformation, but also proteins that act on various components of the tumor stroma. Among the latter is the growth and differentiation factor activin, a member of the transforming growth factor β (TGF-β) family. Activins are homo- or heterodimers composed of two β chains, with activin A (βAβA) being the most abundant and best-characterized variant (Chen et al, 2006). The activity of activins is inhibited through binding to the secreted glycoproteins follistatin or follistatin-related protein in the extracellular environment (Xia & Schneyer, 2009). Activins signal via heterotetrameric receptor complexes consisting of type I and type II receptors, which are transmembrane serine/threonine kinases (Chen et al, 2006).

We previously showed that activin is strongly upregulated in murine and human skin wounds and in human BCCs and SCCs (Hubner et al, 1996; Antsiferova et al, 2011). This is functionally important, since overexpression of activin in keratinocytes of transgenic mice accelerated the wound healing process (Munz et al, 1999), but also promoted skin carcinogenesis and malignant progression of existing tumors in a murine skin cancer model where tumors are induced by a combination of the mutagen 7,12-dimethyl-benzo(a)anthracene (DMBA) and the tumor promoter 12-O-tetradecanoylphorbol-13-acetate (TPA) (Antsiferova et al, 2011).

1 Department of Biology, Institute of Molecular Health Sciences, ETH Zurich, Zurich, Switzerland
2 Department of Dermatology, University Hospital, Zurich, Switzerland
3 Department of Dermatology, University of Lausanne, Lausanne, Switzerland
*Corresponding author. Tel: +41 44 633 3941; Fax: +41 44 633 1147; E-mail: sabine.werner@biol.ethz.ch

Interestingly, when the activin-overexpressing mice were mated with transgenic mice expressing a dominant-negative activin receptor mutant in keratinocytes, tumor development was not inhibited, but rather mildly increased. These findings demonstrate that activin does not promote tumorigenesis via keratinocytes, but rather has a mild tumor-suppressive effect on these cells. By contrast, activin obviously potentiates the oncogenic action of DMBA/TPA by generation of a pro-tumorigenic microenvironment. The latter involves loss of tumor-suppressive γδ T cells in the epidermis and a concomitant increase in cutaneous αβ T cells and Langerhans cells (Antsiferova et al, 2011). However, it is unknown whether these cells are indeed functionally involved in the tumor-promoting effect of activin, and whether activin has additional, and possibly even more important cellular targets. In addition, it remains to be determined whether activin promotes skin tumorigenesis in more physiologically relevant tumor models and whether it is required at an early or late stage of skin cancer development. Most importantly, the molecular targets of activin in the affected cells of the tumor microenvironment remain largely unknown.

Here, we show that activin overexpression is an early event in murine and human skin carcinogenesis and that increased levels of activin strongly promote formation of skin tumors induced by the human papillomavirus 8 (HPV8). We provide functional evidence for a crucial role of macrophages in the pro-tumorigenic effect of activin. Finally, we provide the first genome-wide RNA profiling data of macrophages isolated from pre-tumorigenic lesions, which identified novel activin target genes and revealed that activin induces a pro-tumorigenic macrophage phenotype.

# Results

### Activin A is overexpressed in pre-cancerous skin lesions

To determine whether upregulation of activin expression occurs early during skin cancer development, we analyzed the expression of the activin βA subunit (INHBA) in biopsies of patients with histopathologically well-characterized AK (Dataset EV1). INHBA mRNA levels were two- to eightfold elevated in 7 out of 21 AK biopsies as compared to normal human skin (Fig 1A). This finding suggests that the strong increase in INHBA expression seen in established skin cancers as compared to healthy skin (Antsiferova et al, 2011) is an early event during skin tumorigenesis.

HPV8 has been associated with development of non-melanoma skin cancer (Karagas et al, 2006), and higher HPV8 virus loads were observed in AK (Weissenborn et al, 2005). To determine whether HPV8-induced tumorigenesis is associated with an increase in activin expression, we made use of mice expressing the oncogenes of HPV8 in keratinocytes under the control of the keratin 14 promoter (HPV8/wt mice). These mice spontaneously develop benign skin papillomas (Schaper et al, 2005). There was no significant increase in Inhba mRNA levels in the mildly hyperplastic skin of HPV8/wt compared to control (wt/wt) mice, but a strong upregulation was seen in established papillomas of HPV8/wt mice (Fig 1B). Concomitantly, mRNA levels of follistatin were mildly reduced in the papillomas (Fig EV1A), suggesting that the overexpressed activin is functionally active. Immunostaining identified the tumor cells as well as keratinocytes of the normal epidermis,

endothelial cells and other stromal cells as the sources of activin (Fig 1C–E). This expression pattern is similar to the one observed in human SCCs (Antsiferova et al, 2011) and consistent with the expression of INHBA in tumor cells of AK and SCC patients isolated by laser capture microdissection (Lambert et al, 2014).

### Activin A promotes HPV8-induced skin tumorigenesis in mice

To determine the consequence of activin overexpression in keratinocytes for HPV8-induced skin tumorigenesis, we inter-crossed HPV8 mice with mice overexpressing the activin βA subunit in keratinocytes under the control of the keratin 14 promoter (Act mice) (Munz et al, 1999). While Act mice never developed tumors in the absence of an oncogenic stimulus (Antsiferova et al, 2011), the double-transgenic progeny (HPV8/Act mice) showed a strong increase in the incidence of spontaneously developing skin tumors compared to HPV8/wt animals (Fig 1F), demonstrating that activin potentiates the oncogenic action of the HPV8 transgene. In the HPV8/Act group, the first lesions appeared on the ears at the age of 10 weeks, and this was preceded by progressive epidermal hyperplasia and keratinocyte hyperproliferation (Fig EV1B–E). By week 27, all double-transgenic mice had developed tumors (Fig 1F). In contrast, the first tumors in the HPV8/wt mice only appeared at week 16, and by 80 weeks, only 60% of the animals had developed tumors. The median age of tumor development was 66.5 weeks for HPV8/wt mice and 13 weeks for HPV8/Act mice. Similar to field cancerization in AK patients (Dotto, 2014), several HPV8-induced lesions were usually detected in close proximity in these mice, and they appeared in an area where the epidermis was generally strongly hyperplastic (Fig 1H). Therefore, it was not possible to calculate the tumor multiplicity. The tumors appeared at various anatomical sites (Figs 1G and EV1F), with ear skin, back skin and sites of mechanical irritation (eyelid, snout) being most often affected in both groups. The majority of the lesions were classified as acanthopapillomas or acanthopapillomas with trichoepitheliomatous differentiation. The numbers of both types of lesions were increased in the presence of the INHBA transgene, and there was no major difference in the histopathology between groups (Figs 1H and EV1G). Additionally, six tumors (6.2%) from HPV8/Act, but none from HPV8/wt mice were trichoepitheliomas (Fig EV1G). All lesions analyzed were benign tumors. However, in accordance with the animal welfare regulations, the mice had to be sacrificed when the tumors reached the size of 1 cm$^2$ or when the mice developed more than one tumor with a size of more than 0.5 cm$^2$. Therefore, it may well be that invasive tumors would have developed in older animals and this could have been affected by activin. The strong tumor-promoting effect of activin in the HPV8 model did not result from higher expression of the HPV8 transgene as shown by qRT–PCR analysis of keratinocytes, which had been purified by magnetic cell separation (MACS; Fig EV1H).

### Tumor formation in activin-overexpressing mice correlates with loss of epidermal γδ T cells and accumulation of αβ T cells in the ear skin

To identify the cell types involved in the pro-tumorigenic effect of activin in the HPV8 model, we first analyzed the number of

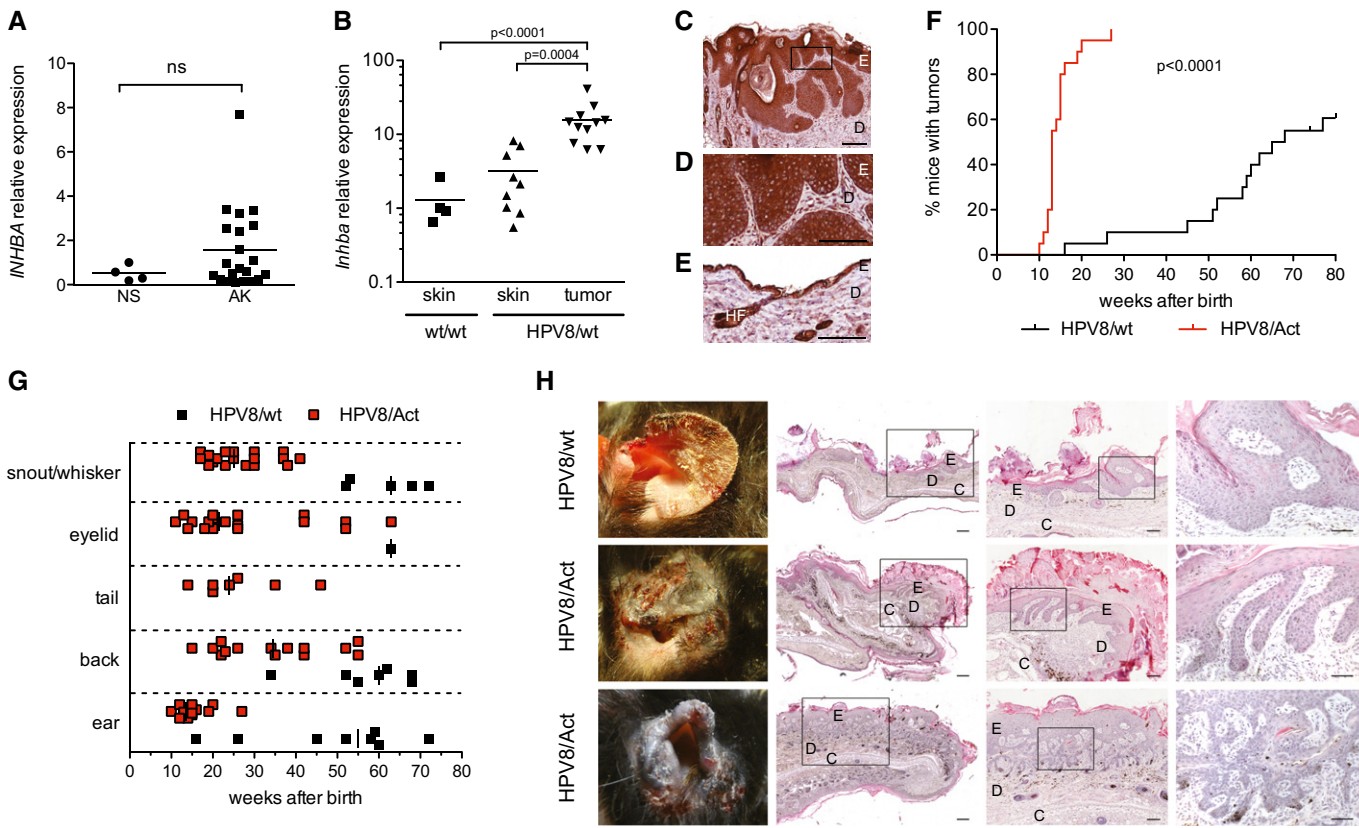

**Figure 1. Activin A is overexpressed in benign skin tumor lesions and promotes HPV8-induced skin tumor development in mice.**

A    qRT–PCR analysis of RNA samples from normal human skin (NS) and AK for activin βA (*INHBA*), normalized to hypoxanthine phosphoribosyltransferase (*HPRT*). Expression levels in one of the normal skin samples were set to 1. Scatter plots are shown. NS, *N* = 4; AK, *N* = 21. ns, non-significant (Mann–Whitney test).

B    qRT–PCR analysis of RNA samples from normal back skin of wild-type mice (*N* = 4, *n* = 4), non-tumorigenic back skin from HPV8/wt mice (*N* = 7, *n* = 9), and papillomas from HPV8/wt transgenic mice (*N* = 8, *n* = 11) for *Inhba*, normalized to glyceraldehyde-3-phosphate dehydrogenase (*Gapdh*). Expression levels in one of the wild-type back skin samples were set to 1. Statistical significance was determined using one-way ANOVA and Bonferroni's multiple comparisons test of log-transformed data. *N*, number of mice; *n*, number of biopsies.

C–E  Sections from a tumor collected from a 64-week-old HPV8/wt mouse (C, D) or from non-tumorigenic skin of 5-week-old HPV8/wt mice (E) stained with an anti-activin A antibody and counterstained with hematoxylin. E, epidermis; D, dermis; HF, hair follicle. Panel (D) shows a higher magnification of the boxed area in (C). Scale bars: 200 μm (C) and 100 μm (D, E).

F    Kinetics of tumor incidence in HPV8/wt and HPV8/Act mice; *N* = 20 mice per genotype. Statistical significance was determined using log-rank (Mantel–Cox) test.

G    Age of the mice when the first tumors appeared on certain body sites. Each dot represents one tumor.

H    Representative macroscopic pictures of ear tumors (left panel) and H&E-stained sections of ear acanthopapillomas (upper and middle panel) or an ear acanthopapilloma with trichoepitheliomatous differentiation (lower panel). Boxed areas are shown at higher magnification in subsequent pictures to the right in each horizontal picture panel. E, epidermis; D, dermis; C, cartilage. Scale bars: 200 μm (left panel), 100 μm (middle panel), or 50 μm (right panel).

---

epidermal T cells expressing the γδ T-cell receptor (TcR). These immune cells, which protect against experimental skin tumorigenesis in mice (Girardi *et al*, 2001), are growth inhibited by activin A *in vitro* and depleted from the skin of Act mice upon DMBA/TPA treatment (Antsiferova *et al*, 2011). Immunostaining revealed a strong reduction in the number of γδ TcR⁺ cells in the ear epidermis of 10-week-old wt/Act and even more in HPV8/Act mice compared to wt/wt or HPV8/wt animals (Fig 2A and B), which correlated with the appearance of tumors in the ear skin at this age. The remaining γδ T cells were activated as shown by the loss of their dendritic morphology (Fig 2A) and by flow cytometry analysis for the activation marker CD69 (Fig EV2A; Appendix Fig S1A). Interestingly, there was no reduction in γδ T cells in the back skin, which correlated with the lack of histological abnormalities in back skin at this age (Fig EV2B and C).

Concomitant with the loss of γδ T cells, there was a significant increase in αβ T cells in the ear, but not in the back epidermis of HPV8/Act and wt/Act mice compared to wt/wt or HPV8/wt animals (Figs 2C and EV2D). Both CD4⁺ and CD8⁺ cells contributed to the increased number of αβ T cells in the dermis and epidermis of the ear skin (Figs 2D and E, and EV2E and F, and Appendix Fig S1B). CD4⁺CD25⁺ TcRβ⁺ cells were particularly abundant in the ear epidermis of HPV8/Act mice (Fig 2F and Appendix Fig S1C), whereas the increase in CD4⁺CD25⁺ TcRβ⁺ cells in the dermis was non-significant (Fig EV2G). This finding indicates accumulation of activated effector or regulatory T cells (Tregs), particularly in the epidermis of HPV8/Act mice. Immunofluorescence for the Treg-specific transcription factor Foxp3 indeed revealed a striking accumulation of these immunosuppressive cells in the activin-overexpressing mice, in particular in the presence of the activin

transgene (Fig 2G). The increased number of αβ T cells most likely resulted from the upregulation of different T-cell chemokines, in particular chemokine ligand 22 (*Ccl22*) (Fig EV2H–J).

## Depletion of CD4⁺ T cells does not reduce the tumor-promoting effect of activin

As CD4⁺ T cells enhanced neoplastic progression in HPV16-induced skin cancer (Daniel *et al*, 2003) and promoted skin

tumorigenesis in DMBA/TPA- and UV-induced skin cancer models (Yusuf *et al*, 2008; Nasti *et al*, 2011), and because of the increase in CD4-positive Tregs, we intercrossed Act and HPV8 with CD4-deficient mice and analyzed spontaneous tumorigenesis in the single, double, and triple mutant progeny. Surprisingly, depletion of CD4⁺ cells caused a slight, albeit non-significant increase in tumor incidence in wt/HPV8 mice, indicating that CD4⁺ T cells are rather tumor-suppressive in the HPV8 model under our experimental conditions. Most importantly, depletion

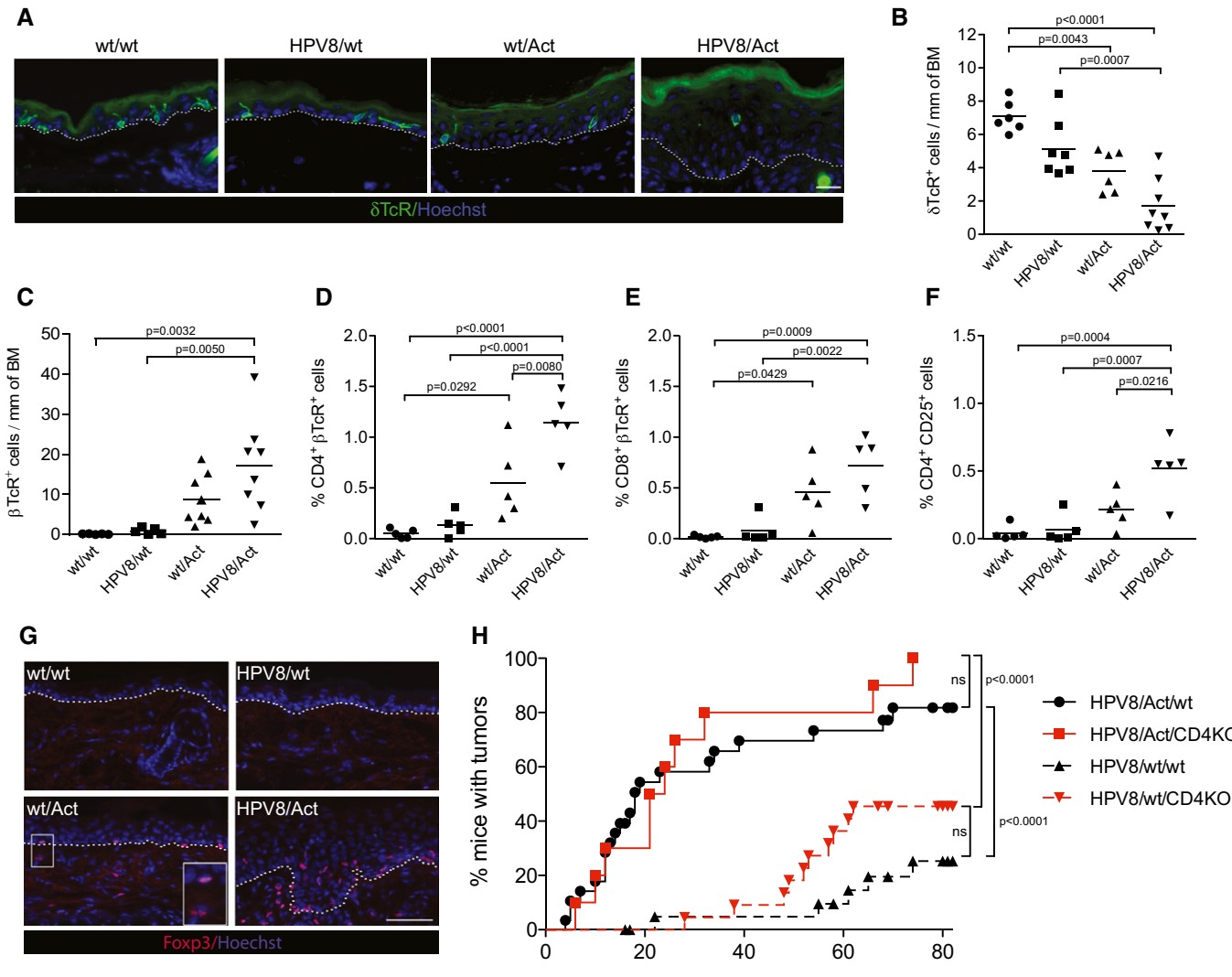

**Figure 2.  Depletion of epidermal γδ T cells and accumulation of αβ T cells in the ear skin of activin-overexpressing mice.**

A, B    Representative pictures of ear sections from 10-week-old mice stained for δTcR (green) and counterstained with Hoechst (blue) (A), and quantification of δTcR⁺ cells in the ear epidermis (B). Dotted lines indicate the epidermal–dermal border. Scale bar: 20 μm. *N* = 6 wt/wt mice, *N* = 7 HPV8/wt mice, *N* = 6 wt/Act mice, and *N* = 8 HPV8/Act mice.

C    Quantification of βTcR⁺ cells in ear skin cryosections from 10-week-old wt/wt (*N* = 5), HPV8/wt (*N* = 5), wt/Act (*N* = 8) or HPV8/Act (*N* = 8) mice.

D–F    Percentage of viable CD4⁺ βTcR⁺ (D), CD8⁺ βTcR⁺ (E) or CD4⁺CD25⁺ (F) cells among all epidermal cells analyzed by flow cytometry. Results from five experiments with ears pooled from at least three mice per genotype are shown. Representative flow cytometry scatter plots are shown in Appendix Fig S1A–C.

G    Representative pictures of ear cryosections from 10-week-old mice stained with antibodies against Foxp3 (red) and counterstained with Hoechst (blue). Insert shows higher magnification of the Foxp3-positive cells in wt/Act skin. Dotted lines indicate the epidermal–dermal border. Scale bar: 50 μm.

H    Kinetics of tumor incidence in HPV8/Act/wt (*N* = 29), HPV8/Act/CD4KO (*N* = 10), HPV8/wt/wt (*N* = 23), and HPV8/wt/CD4KO (*N* = 22) mice per genotype. Statistical significance was determined using the log-rank (Mantel–Cox) test; ns, non-significant.

Data information: Statistical significance was determined using one-way ANOVA and Bonferroni's multiple comparisons test (B–F).

of CD4[+] T cells did not abolish the tumor-promoting effect of activin (Fig 2H).

### Activin A increases the number of macrophages in the ear skin

The striking increase in dermal cellularity in pre-cancerous ear skin of HPV8/Act mice prompted us to determine the number of macrophages. Indeed, immunofluorescence staining revealed a strong accumulation of CD206-positive macrophages in the dermis of the hyperplastic ear skin of wt/Act and HPV8/Act mice, but not of HPV8/wt mice (Fig 3A and B). This was confirmed by flow cytometry analysis using antibodies against the macrophage markers MerTK and CD64 (Gautier *et al*, 2012) (Fig 3C and Appendix Fig S1D). The increased numbers of macrophages in the skin of Act mice, even in the absence of the *HPV8* transgene, demonstrate that activin rather than the *HPV8* oncogenes is responsible for this accumulation. The important role of activin in macrophage accumulation was confirmed in the DMBA/TPA model, where the number of

macrophages was also significantly increased in Act compared to wt mice after eight TPA treatments (Fig 3D).

As a substantial fraction of dermal macrophages is continuously replenished by blood-borne precursors (Tamoutounour *et al*, 2013), we investigated the origin of skin macrophages that accumulate in the ears of activin-overexpressing mice by mating of K14-Act mice with CCR2-eCFP-DTR mice (Hohl *et al*, 2009). These mice express enhanced cyan fluorescent protein (CFP) and the diphtheria toxin receptor (DTR) under control of the C-C chemokine receptor 2 (*CCR2*) promoter and allow monitoring of CCR2[+] cells by analysis of CFP fluorescence. They also allow depletion of these cells by systemic treatment of the animals with DT. Since CCR2 is required for emigration of monocytes from the bone marrow (Serbina & Pamer, 2006), depletion of this population allowed us to determine whether the macrophages that accumulate in the skin of Act mice are derived from the bone marrow. Depletion of CCR2[+] cells by injection of DT every second day during 1 week indeed dramatically reduced the number of MerTK[+]CD64[+]

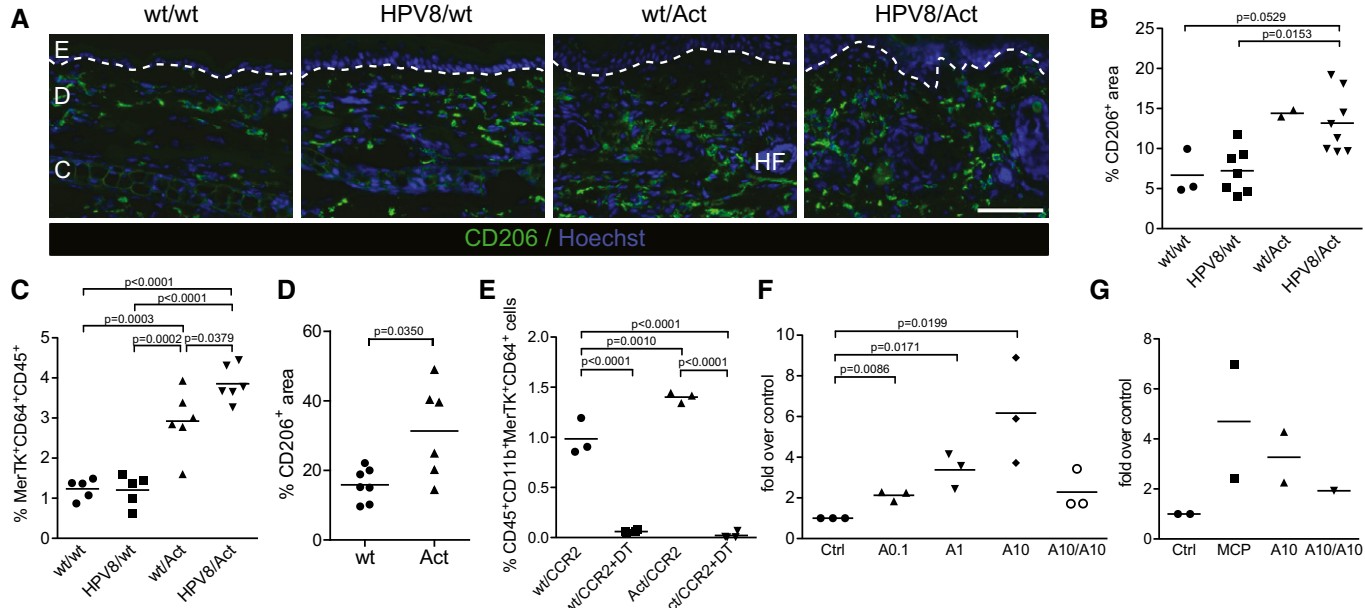

**Figure 3.  Activin promotes migration of CCR2[+] macrophage precursors to the skin.**

A, B    Representative pictures of ear skin cryosections from 10- to 12-week-old mice stained with anti-CD206 antibodies (green) and counterstained with Hoechst (blue) (A), and quantification of CD206[+] area as a percentage of total dermal area (B) in wt/wt (*N* = 3), HPV8/wt (*N* = 7), wt/Act (*N* = 2), or HPV8/Act (*N* = 8) mice. The basement membrane is indicated with a white dotted line.

C    Abundance of MerTK[+]CD64[+] macrophages analyzed by flow cytometry of normal or pre-cancerous ear skin from 12- to 16-week-old mice. *N* = 5–6. Representative flow cytometry scatter plots are shown in Appendix Fig S1D.

D    Quantification of CD206[+] area as a percentage of total dermal area in wt (*N* = 7) and Act (*N* = 6) mice in back skin treated 8× with TPA (Antsiferova *et al*, 2011).

E    Abundance of MerTK[+]CD64[+] macrophages analyzed by flow cytometry of pre-tumorigenic ear skin from 12-week-old mice expressing the CCR2-CFP-DTR transgene (wt/CCR2) or double-transgenic animals (Act/CCR2), untreated or treated with diphtheria toxin (+DT) three times every 48 h and analyzed 24 h after the last injection. Percentage of CD45[+]CD11b[+]MerTK[+]CD64[+] macrophages among live cells is shown. *N* = 3 control wt/CCR2 and Act/CCR2 mice; *N* = 4 DT-injected wt/CCR2 and Act/CCR2 mice. Representative flow cytometry scatter plots are shown in Appendix Fig S1E.

F, G    Inflammatory thioglycollate-elicited peritoneal macrophages (F) or mouse peripheral blood monocytes (G) were isolated from wt mice and allowed to migrate for 24 h in a Transwell assay. Activin A 0.1 ng/ml (A0.1), 1 ng/ml (A1), or 10 ng/ml (A10) was added to the lower well. Alternatively, 10 ng/ml was added to the lower well and the insert (A10/A10). Random migration to medium (ctrl) was set to 1 for each mouse; fold over control is shown. Log-transformed data were used for statistical analysis. Results in (F) are representative of four independent experiments performed with eight mice in total (results obtained in one experiment performed with three mice are shown). Results in (G) are combined from two independent experiments performed with monocytes isolated from pooled blood of at least 13 wt mice.

Data information: Statistical significance was determined using one-way ANOVA and Bonferroni's multiple comparisons test (B–E) or one-sample *t*-test after log-transformation (F).

cells in the ear skin of wt/CCR2 and Act/CCR2 mice (Fig 3E and Appendix Fig S1E), indicating that the accumulation of macrophages in the skin is a result of increased influx of $CCR2^+$ monocytes. This is most likely a direct effect of activin as shown in Transwell migration assays where recombinant activin A dose-dependently increased the migration of monocyte-derived inflammatory peritoneal macrophages (Fig 3F). However, when activin A was present in the lower and the upper well of the Transwell assay plate, macrophage migration was only marginal, indicating that an activin gradient induces directional migration of macrophages. Consistent with this finding, activin A also attracted FACS-sorted blood monocytes (Fig 3G).

**Activin induces a pro-migratory and tumor-promoting phenotype in macrophages *in vivo* and *in vitro***

To determine the *in vivo* relevance of this finding and the potential relevance for skin tumorigenesis, we isolated macrophages from the pre-tumorigenic ear skin of wt/wt, HPV8/wt, wt/Act, and HPV8/Act mice by FACS, and analyzed their expression profile by RNA sequencing. This experiment was performed after transfer of our mice to a new animal facility where the development of tumors was delayed in HPV8/Act and in particular in HPV8/wt mice compared to the old facility (Fig EV3A), most likely due to minor alterations in the microbiome and/or environmental conditions. The new environment affected the onset of tumor formation, but not the strong pro-tumorigenic effect of activin and the effect of activin on immune cells. Therefore, we used pre-tumorigenic skin of 13- to 15-week-old wt/wt, HPV8/wt, wt/Act, and HPV8/Act mice for the FACS-sorting experiment (Fig EV3B), and we sequenced RNA samples from three pools of 3–6 mice per genotype (Fig 4A). Each mouse was carefully analyzed for any signs of tumor formation, and only mice without

tumors were used for the sorting experiment. We sorted $F4/80^+/CD11b^+$ cells to obtain a broad coverage of the myeloid cell population and to obtain a sufficient number of sorted cells from ear skin, which was particularly difficult in mice lacking the activin transgene (Fig 4B). The sorting confirmed that activin, but not the *HPV*8 transgene, strongly enhances the number of macrophages in the skin (Fig 4B).

The *HPV8* transgene alone also did not affect the gene expression pattern in macrophages, but activin overexpression had a dramatic effect on the expression profile of $F4/80^+CD11b^+$ cells, both in the presence or absence of the *HPV8* transgene (Fig EV3C). Thus, 107 genes were differentially regulated (FDR < 0.05, |log$_2$ratio| > 1) in wt/Act vs. wt/wt mice, 80 genes in HPV8/Act vs. HPV8/wt mice, and 128 genes in HPV8/Act vs. wt/wt mice (Fig EV3C and D). Complete lists of differentially regulated genes from different genotype comparisons are presented in Dataset EV2. Analysis of the expression data confirmed that mouse skin macrophages express activin receptors *Acvr1, Acvr1b, Acvr2a,* and *Acvr2b* (Fig EV3E), explaining their responsiveness to activin A.

We then used Ingenuity Pathway Analysis (IPA) to determine whether genes associated with certain functions or diseases are enriched in the activin-regulated genes using all three genotype comparisons: wt/Act vs. wt/wt; HPV8/Act vs. HPV8/wt; HPV8/Act vs. wt/wt. Interestingly, genes associated with "cell movement" and "activation of mononuclear leukocytes", "adhesion of phagocytes", and "vasculogenesis" were significantly enriched, and their regulation by activin corresponded with the predicted activation of these functions in the sorted skin macrophages (Fig 4C). With regard to disease association, genes annotated with "cancer" and—more specific—"skin tumor" were significantly enriched and predicted to be activated in the HPV8/Act vs. control, but not in the wt/Act vs.

---

**Figure 4.  Activin reprograms macrophages toward a TAM-like phenotype.**

A  Scheme of the experimental setup. Pre-tumorigenic ear skin from 3 to 6 mice per genotype was pooled, and macrophages were enriched by FACS sorting of $F4/80^+CD11b^+$ cells and subjected to RNA sequencing. Each replicate represents one of three independent FACS experiments (*n* = 3).

B  Analysis of FACS data of sorted macrophages shows the relative percentage of $CD45^+CD11b^+F4/80^+$ cells in each genotype and the absolute number of sorted cells used for RNA sequencing. Representative flow cytometry gating strategy for sorting is shown in Fig EV3B. N = 5–9. Statistical significance was determined using one-way ANOVA and Bonferroni's multiple comparisons test.

C  Differential expression of genes was analyzed using the edgeR software package. IPA was performed with filtered lists of up- and downregulated genes that exhibited an FDR value of < 0.1 in different genotype comparisons (wt/Act vs. wt/wt; HPV8/Act vs. HPV8/wt; HPV8/Act vs. wt/wt). The number of differentially expressed genes used for analysis is indicated. Data are presented as pseudo-heatmap with magnitudes of activation *Z*-scores color-coded as indicated in the legend; all non-gray comparisons are −log$_{10}$(*P*-value) > 2.5 unless otherwise indicated.

D  Gene set enrichment analysis (GSEA) was performed to compare transcriptomes of genotype comparisons (wt/Act vs. wt/wt; HPV8/Act vs. HPV8/wt; HPV8/Act vs. wt/wt, [all]/Act vs. [all]/wt) to three published datasets of tumor-associated macrophages (TAM) (Ojalvo *et al*, 2009; Franklin *et al*, 2014; Galletti *et al*, 2016). Gene sets are composed of top genes upregulated (TAM UP) or downregulated (TAM DOWN) in TAM vs. normal tissue macrophages. Positive normalized enrichment score (NES) corresponds to enrichment of gene set in activin-upregulated genes; negative NES corresponds to enrichment in activin-downregulated genes. Data are presented as pseudo-heatmap with magnitudes of NES color-coded as indicated in the legend; all non-gray comparisons are FDR < 0.10.

E  Differential GSEA was performed on combined genotype comparison [all]/Act vs. [all]/wt and two datasets of human AK vs. normal skin (GSE2503; Nindl *et al*, 2006) and GSE63107 to compare these transcriptomes to gene sets derived from top upregulated genes in these datasets (top); and gene sets derived from top upregulated genes in datasets of differential human macrophage activation (IFN-γ, TNF-α, LPS, IL-4 vs. non-treated) and selected human myeloid/lymphoid cell-type-specific gene expression profiles (macrophage vs. monocyte, dendritic cell, and T cell) (Xue *et al*, 2014; bottom). Activin target genes enriched in the AK transcriptomes are shown (right). Data are presented as pseudo-heatmaps with magnitudes of normalized enrichment scores (NES) and log$_2$(fold change) color-coded as indicated in the respective legends; all non-gray comparisons are FDR < 0.05 for NES and *P* < 0.10 for fold change.

F  Venn diagram showing overlapping differentially expressed genes (FDR ≤ 0.05, |log$_2$ratio| ≥ 1) in different genotype comparisons.

G  Heatmap of 28 differentially expressed genes (FDR ≤ 0.05, |log$_2$ratio| ≥ 1) overlapping in all three comparisons (wt/Act vs. wt/wt mice, HPV8/Act vs. HPV8/wt and HPV8/Act vs. wt/wt). Genes with a published pro-tumorigenic function are highlighted in red.

Data information: Complete lists of differentially regulated genes from different genotype comparisons, including corresponding fold change and exact FDR values, are presented in Dataset EV2. Complete list of diseases and functions identified by IPA for each genotype comparison, including exact *P*-values, is shown in Dataset EV3. Complete original and filtered gene sets, ranked gene lists used for the GSEA, and original GSEA reports, including exact FDR and *P*-values, can be found in Dataset EV4.

wt/wt comparison. The "cell proliferation of SCC cell lines" function was both enriched and activated in all comparisons. The full list of diseases and functions identified by IPA for each genotype comparison can be found in Dataset EV3.

We next used gene set enrichment analysis (GSEA) (Subramanian *et al*, 2005) to compare activin- and HPV8-dependent gene expression profiles of sorted macrophages with published transcriptomes of tumor-associated macrophages (TAMs). Remarkably, genes regulated by activin in macrophages showed a significant positive correlation with genes that were shown to be upregulated

in murine mammary TAMs in two independent studies (Ojalvo *et al*, 2009; Franklin *et al*, 2014) and also in bone marrow TAMs from mice with leukemia (Galletti *et al*, 2016) (Fig 4D). In contrast, activin-regulated genes were negatively correlated with the genes downregulated in TAMs and thus upregulated in normal tissue macrophages.

To determine the potential relevance of our mouse RNA-seq data for human AK, we made use of two sets of published RNA profiling data from normal skin and AK samples (GSE63107; Nindl *et al*, 2006). We performed GSEA to assess the level of overall gene

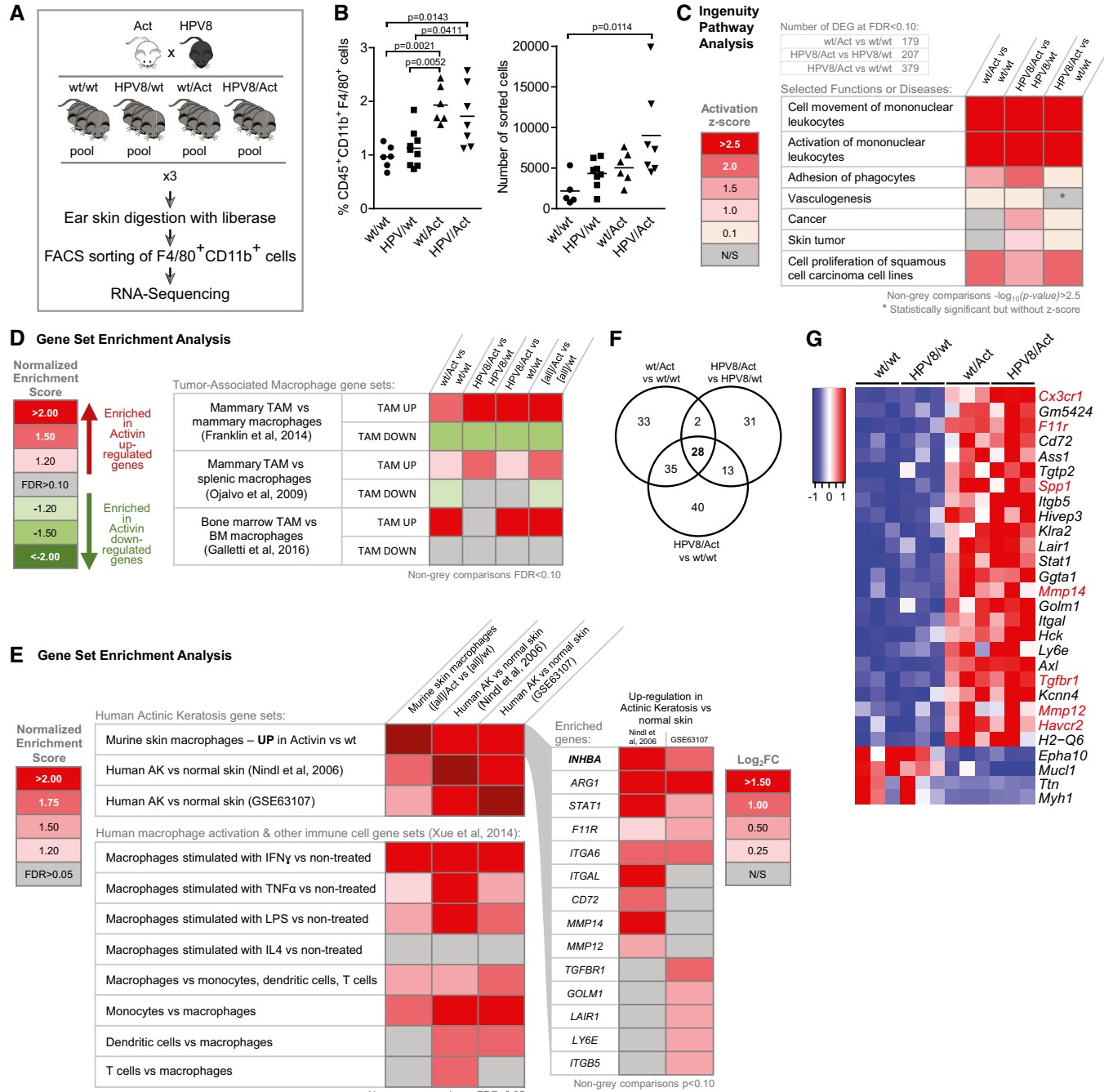

**Figure 4.**

expression similarities between murine macrophages exposed to the transgene-derived activin and total AK samples. Importantly, many of the genes that were upregulated in mouse macrophages by activin *in vivo* were strongly enriched in both AK datasets. Likewise, genes upregulated in both AK datasets were enriched in activin-stimulated murine macrophages (Fig 4E, top panel). Further analysis of these data confirmed the upregulation of *INHBA* in AK samples, together with several activin target genes in macrophages that we had identified, including *ARG1, F11R, MMP12, MMP14, TGFBR1,* and others (Fig 4E, right panel).

We also performed comparative GSEA on several relevant gene sets derived from expression profiling experiments of *in vitro* stimulated human macrophages and a subset of myeloid and lymphoid cells in order to assess the potential contribution of these cell types to the overall gene expression of AK samples and to compare them to the macrophage phenotype seen in Act mice. Intriguingly, the gene expression profile of classically activated macrophages (IFN-γ treated; Xue *et al*, 2014) was strongly enriched in our activin-stimulated macrophages and in both AK datasets, while that of alternatively activated macrophages (IL-4 treated) was not (Fig 4E, bottom). GSEA revealed a monocyte gene signature in activin-exposed macrophages that is also present in both AK datasets in addition to other myeloid and lymphoid cell signatures, due to the complex nature of the clinical samples. Overall, these results suggest that monocyte recruitment and stimulation of macrophages by activin may contribute significantly to the gene expression profile and pathophysiology of human AK.

We then concentrated on the 28 genes that were differentially regulated in all three genotype comparisons (24 up, four down; FDR ≤ 0.05, |log$_2$ratio| 1) (Fig 4F and G). A literature search for these genes allowed us to assign them to the functional categories "adhesion/migration", "cancer-association", "angiogenesis", "anti-microbial/effector function", and "immunoregulation", thus mirroring the results from the IPA analysis (Appendix Table S1).

Consistent with the migratory phenotype of macrophages in activin-transgenic mice, a class of genes that was strongly affected by activin encodes proteins associated with cell migration. Among them is CX3CR1, the fractalkine receptor that is expressed among other cells on blood monocytes and is thought to mediate their recruitment to non-inflamed tissues (Geissmann *et al*, 2003). We confirmed the upregulation of *Cx3cr1* mRNA in a separate set of biological replicates (Appendix Fig S2A), and its expression was strongly induced by activin A in peritoneal macrophages *in vitro* (Fig 5A). Overexpression of several adhesion molecules (junctional adhesion molecule A (*F11r*), integrin beta 5 (*Itgb5*), integrin alpha L, (*Itgal*)), and of other proteins associated with cell migration (Golgi membrane protein 1 (*Golm1*) and haematopoietic cell kinase (*Hck*)), was also observed in the presence of the activin transgene (Appendix Table S1).

Most interestingly, 8 out of 26 (31%) of the genes significantly upregulated in all three comparisons have a documented pro-tumorigenic function, are expressed by TAMs, or serve as a prognostic marker of advanced/malignant cancers (Appendix Table S1, highlighted in red in Fig 4G). One of them, *Spp1*, encodes osteopontin, which promotes fibrotic processes such as scarring of skin wounds (Mori *et al*, 2008) and tumorigenesis (Shevde & Samant, 2014). It was more than fourfold upregulated in macrophages isolated from activin-overexpressing mice (Fig EV3D and E), and its expression was strongly induced by activin A in peritoneal macrophages *in vitro* (Fig 5B). We confirmed the increased expression of osteopontin by Western blot analysis of total ear lysates (Fig 5C and D) and by immunofluorescence staining of ear skin sections, where most of the staining co-localized with CD68$^+$ macrophages (Fig 5E).

Although not significantly regulated in all genotype comparisons, arginase 1 (*Arg1*), which promotes skin tumorigenesis in mice (Weber *et al*, 2016), was also strongly upregulated in the presence of the Act transgene (Fig EV3D and E, and Appendix Fig S2B). This is most likely a direct effect of activin, since activin A induced the expression of *Arg1* in resident and inflammatory thioglycollate-elicited peritoneal macrophages (Fig 5F).

---

**Figure 5. Activin induces expression of pro-tumorigenic genes in macrophages, associated with a proteolytic microenvironment and vascular remodeling.**

A, B   Expression of *Cx3cr1* (A) and *Spp1* (B) relative to *Rps29* was analyzed by qRT–PCR in peritoneal macrophages, resident or inflammatory, treated *in vitro* with 10 ng/ml of activin A (Act) or untreated (Ctrl, arbitrarily set to 1). Results are combined from 2 to 5 independent experiments performed with cells pooled from at least five wt mice.

C, D   Western blot analysis of total ear protein lysates of 12- to 14-week-old mice for osteopontin (OPN) and GAPDH (C). Each lane represents an individual mouse (*n* = 3). (D) Band intensities were quantified and normalized to GAPDH. The expression level in one of the wt mice was arbitrarily set to 1.

E      Ear skin sections from 10-week-old mice stained with anti-CD68 (red) or anti-osteopontin antibody (green) and counterstained with Hoechst (blue). Osteopontin-positive macrophages in wt/Act and HPV8/Act mice are indicated with arrows. Representative of *N* = 2 wt/wt and *N* = 3 HPV8/wt, wt/Act and HPV8/Act mice. Boxed areas in top panel are shown at higher magnification in bottom panel. Scale bars: 100 μm (top) or 20 μm (bottom).

F      Expression of *Arg1* relative to *Rps29* was analyzed by qRT–PCR in peritoneal macrophages, resident or inflammatory, treated *in vitro* with 10 ng/ml of activin A (Act) or untreated (Ctrl, arbitrarily set to 1). Results are combined from 2 to 5 independent experiments performed with cells pooled from at least five wt mice.

G      *In situ* gelatin zymography (top panel) and methylene blue staining (bottom panel) of the ear skin from 10-week-old mice. Representative images of 2–3 mice per genotype are shown. Scale bar: 500 μm.

H, I   Gelatin zymography of ear skin lysates from 10-week-old mice. Inverted image of a representative gel with samples from two (wt/wt) or three (HPV8/wt, wt/Act, HPV8/Act) mice per genotype is shown in (H), quantification of the results obtained in two independent experiments is shown in (I). *N* = 5 wt/wt, *N* = 6 HPV8/wt, wt/Act, and HPV8/Act mice.

J      Representative pictures of ear skin from 10-week-old mice stained for MECA32 (red), LYVE1 (green), and counterstained with Hoechst (blue). Boxed areas in top panel are shown at higher magnification in bottom panel. Scale bars: 100 μm (top) or 20 μm (bottom).

K, L   Quantification of MECA32$^+$ area (K) and average distance between MECA32$^+$ vessels and basement membrane (BM) (L).

Data information: Statistical significance was determined using one-sample *t*-test of log-transformed data (A, B, F), one-way ANOVA, and Bonferroni's multiple comparisons test of log-transformed data (D) or one-way ANOVA and Bonferroni's multiple comparisons test (I, K, L).
Source data are available online for this figure.

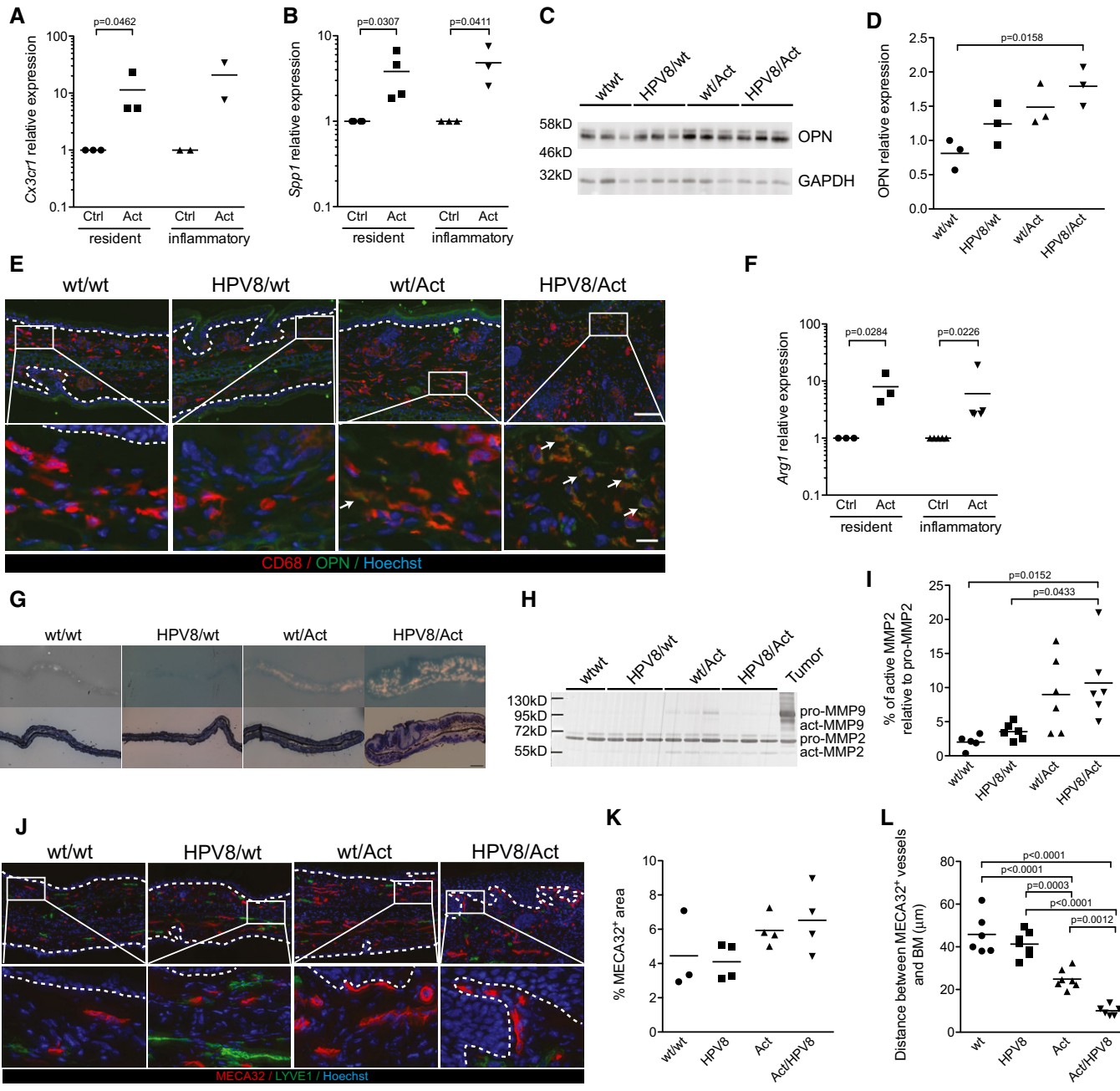

**Figure 5.**

Skin macrophages of activin-transgenic mice also showed a strong overexpression of the gene encoding matrix metalloproteinase (MMP)-14, an activator of MMP-2 (Sato *et al*, 1994) (Figs 4G and EV3D and E). This is likely to be functionally relevant as determined by *in situ* zymography. There was little or no gelatinolytic activity in ear skin of 10-week-old wt/wt and HPV8/wt mice, whereas the dermis of wt/Act, and especially of HPV8/Act animals exhibited strong gelatinolytic activity (Fig 5G), a phenomenon that is often associated with a pro-tumorigenic microenvironment (Kessenbrock *et al*, 2010). In-gel gelatin zymography revealed that this increase mainly results from MMP-2 activity, whereas pro- and active MMP-9 were hardly detectable (Fig 5H and I). Macrophages

from activin-overexpressing mice also showed a strong upregulation of *Mmp12* (Figs 4G and EV3E), and a moderate increase in *Mmp13* expression (Fig EV3E).

Given the important functions of MMP-13, -14, and -2, and of osteopontin in tumor angiogenesis (Basile *et al*, 2007; Kudo *et al*, 2012; Kale *et al*, 2014), we analyzed the area and the size of the blood vessels, but did not find significant changes in activin-overexpressing mice (Fig 5J and K, and data not shown). An obvious difference, however, was the accumulation of vessels in the upper dermis in these mice, as evidenced by the significantly reduced distance between MECA32$^+$ vessels and the basement membrane (Fig 5J and L). This finding suggests that the activin produced by

keratinocytes directly or indirectly induced angiogenesis and vascular remodeling in the underlying dermis.

Taken together, when overexpressed in the skin, activin not only attracts monocytes to the dermis, but also affects their differentiation, thereby inducing an expression profile that promotes migration, angiogenesis, and ultimately tumorigenesis.

### Macrophage depletion delays spontaneous tumorigenesis in Act/HPV8 mice

Finally, we determined whether the pro-tumorigenic effect of activin is indeed macrophage-dependent. Therefore, we depleted these cells by injecting HPV8/Act mice with an antibody against the receptor of colony-stimulating factor 1 (CSF1R) twice per week prior to the first manifestation of tumors (30–37-day-old mice) during a 5-week period. For this purpose, we mated Act mice with HPV8 mice in a FVB1 background, since tumors develop earlier in this mouse strain compared to C57BL/6 mice (Schaper et al, 2005). Macrophage depletion significantly delayed the spontaneous tumorigenesis in HPV8/Act mice as determined by macroscopic analysis of both ears (Fig 6A). This was confirmed by histopathological analysis of the ear skin samples collected 3–4 days after the treatment. In the ear sections that were analyzed (one section from each ear), we detected acanthopapillomas in 4 out of 10 HPV8/Act mice treated with control IgG, whereas we only observed intra-epithelial neoplasia in the sections from the 10 macrophage-depleted animals. Immunostaining confirmed the strong reduction in the number of CD68$^+$ or CD206$^+$ cells in the ear skin compared to mice injected with control rat IgG (Fig 6B and C). Thus, the strong increase in tumor formation in Act mice is to a large extent dependent on macrophages, which accumulate in the skin and acquire a pro-tumorigenic phenotype in the presence of activin. The effect of CSF1R antibody treatment on tumorigenesis in HPV8/wt mice could not be determined, since these mice do not develop tumors within this time frame (Fig EV3A) and since multiple injections per week over many months were not possible because of animal welfare reasons. Therefore, we cannot exclude the possibility that tumor formation in HPV8/wt mice is also at least in part dependent on macrophages, although these cells did not accumulate in the pre-tumorigenic skin of HPV8/wt mice and their gene expression pattern was not affected by the HPV8 transgene.

The reduced tumor formation upon macrophage depletion in HPV8/Act mice was associated with a significantly reduced proliferation rate of keratinocytes (Fig 6D and E). Consistent with this finding, depletion of CCR2$^+$ cells from Act/CCR2 mice using DT (see Fig 3F) also reduced keratinocyte proliferation (Appendix Fig S3A and B). Notably, macrophage depletion was accompanied by a strong downregulation of Mmp12 mRNA (Fig 6F). This is consistent with the efficient depletion of macrophages, since MMP-12, which is also known as macrophage metalloelastase, is predominantly expressed in this cell type (Qu et al, 2011). In addition, we found a significant reduction in the area of CD31$^+$LYVE1$^-$ blood vessels (Fig 6G and H) and a decreased number of blood vessels (Fig 6G and I). These findings reveal that activin promotes skin tumorigenesis at least in part via macrophages. They also strongly suggest that activin promotes angiogenesis, proteolytic activity, and keratinocyte proliferation at least in part via its effect on macrophages.

## Discussion

Here, we show a remarkable pro-tumorigenic effect of activin in the early phase of skin tumorigenesis, which is mediated via activin-induced attraction of monocytes and their differentiation into a TAM-like phenotype. The early upregulation of activin during epithelial skin tumorigenesis in mice and humans may be a consequence of inflammation, since activin A expression is induced in cultured keratinocytes by pro-inflammatory cytokines (Hubner & Werner, 1996) and since AK lesions and also papillomas in HPV8-transgenic mice are characterized by an inflammatory infiltrate (Massone & Cerroni, 2015; Rolfs et al, 2015; M. Antsiferova and S. Werner, unpublished data). Thus, activin seems to be particularly relevant at the stage when benign tumors appear and further progress. By contrast, there is little inflammation in the pre-tumorigenic, but already mildly hyperplastic skin of HPV8-transgenic mice (Rolfs et al, 2015), including a lack of macrophage accumulation, which correlates with the lack of activin upregulation at this very early stage.

The inflammation-induced increase in activin expression is functionally important, since activin overexpression promoted the development of skin papillomas in the DMBA/TPA and also in the HPV8 skin tumorigenesis model. In the future, it will be interesting to determine whether overexpression of activin promotes malignant conversion of AK lesions and could thus act as a biomarker for lesions, which have a higher probability to progress to SCC. In support of this hypothesis, we previously showed in the DMBA/TPA model that activin promotes malignant conversion of existing skin tumors (Antsiferova et al, 2011). However, a recent study demonstrated that only 2.57% of AKs progress to SCC within a 4-year interval (Criscione et al, 2009), suggesting that overexpression of activin alone is not sufficient for progression into SCC. Rather, the malignant progression likely requires additional mutations and/or epigenetic alterations in keratinocytes. Nevertheless, activin overexpression may be a relevant diagnostic parameter. Upon progression of AK to SCC, a further increase in activin expression occurs (Antsiferova et al, 2011; Lambert et al, 2014), suggesting that activin not only promotes malignant progression of early lesions, but also growth and potentially malignant progression of established SCCs.

Similar to the DMBA/TPA model, activin overexpression also reduced the number of tumor-suppressive γδ T cells and increased the number of tumor-promotive Tregs in ear skin of HPV8-transgenic mice, which correlated with the appearance of skin tumors. However, genetic depletion of CD4$^+$ T cells did not rescue the pro-tumorigenic effect of activin. It even slightly increased the tumor incidence in wt/HPV8 mice, a finding that contradicts the tumor-promotive role of CD4$^+$ T cells in a HPV16 skin cancer model (Daniel et al, 2003). The difference may be due to the different viral transgenes and/or different hygiene conditions, since CD4$^+$ T cells from K14-HPV16 mice showed reactivity to Staphylococcus aureus antigens and induced neutrophil infiltration, resulting in tumor promotion. Unfortunately, interpretation of the result obtained with the CD4-deficient mice is problematic, since the CD4$^+$ T-cell population not only includes Tregs, but also other types of T cells that may induce an immune response against the tumor cells. Loss of both populations may explain why we did not observe a major effect on tumorigenesis in the presence of high activin levels. In the future, it

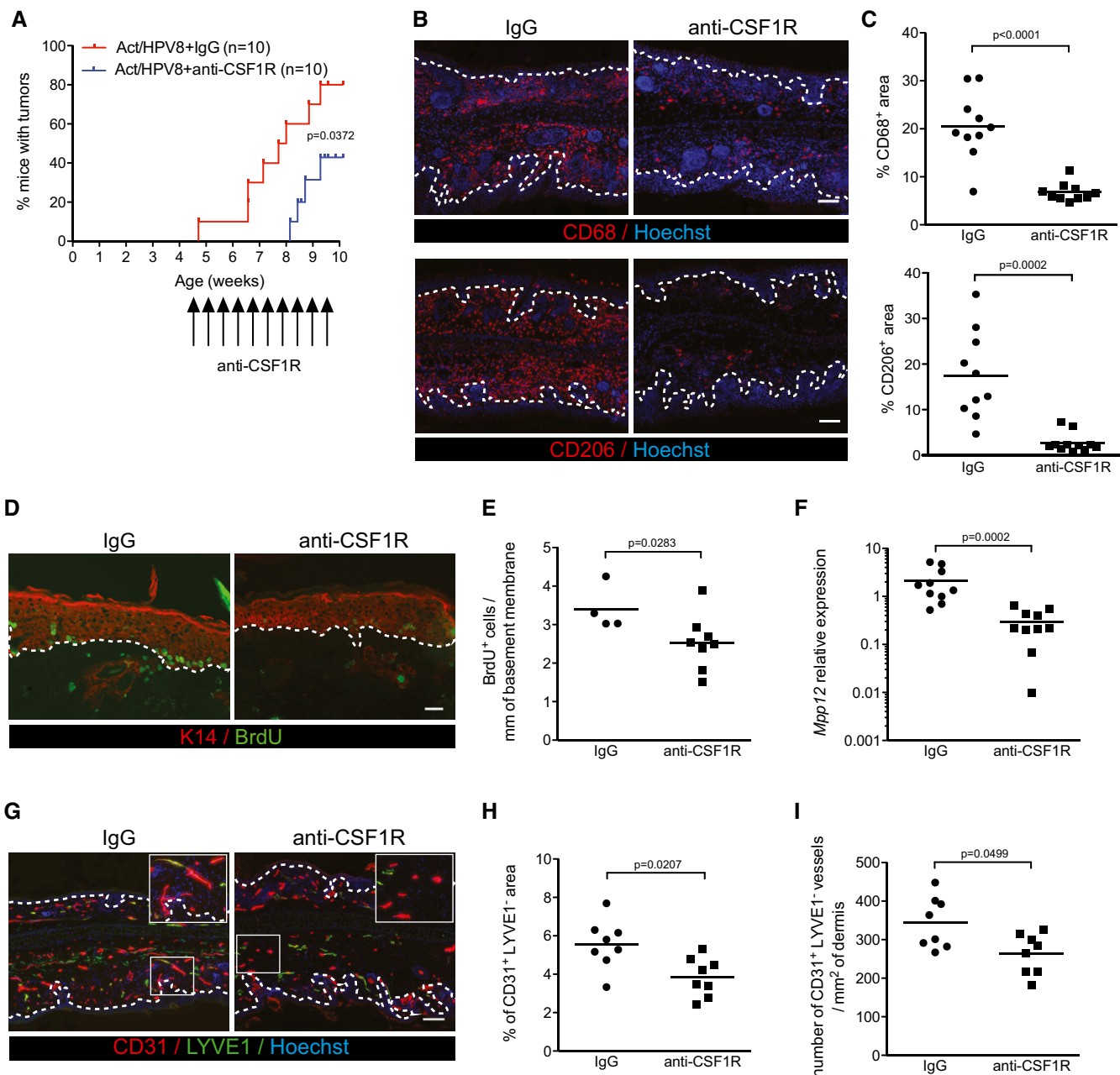

**Figure 6.  Depletion of macrophages delays spontaneous tumorigenesis in HPV8/Act mice by reducing keratinocyte proliferation and angiogenesis.**

A    Kinetics of tumor incidence in HPV8/Act mice treated for 5 weeks with 50 mg/kg anti-CSF1R antibody (AFS98) or with control rat IgG. *N* = 10 mice per treatment group.

B    Representative pictures of anti-CD68- or anti-CD206-stained cryosections of ear skin from 10-week-old mice treated for 5 weeks with anti-CSF1R antibody or control rat IgG. Scale bar: 100 μm.

C    Quantification of the CD68$^+$ or CD206$^+$ area relative to total dermis area. *N* = 10 mice per treatment group.

D    Representative pictures of anti-BrdU (green) and anti-K14 (red) stained cryosections of ear skin from 10-week-old mice treated for 5 weeks with anti-CSF1R antibody or control rat IgG. Scale bar: 50 μm.

E    Quantification of BrdU$^+$ keratinocytes per mm of basement membrane. *N* = 8 anti-CSF1R-injected mice; *N* = 4 rat IgG-injected mice.

F    Expression of *Mmp12* relative to *Rps29* was analyzed by qRT–PCR in total ear skin from mice treated with anti-CSF1R antibody or with rat IgG. Expression in one of the IgG-injected mice was set to 1. *N* = 10 per group.

G    Representative pictures of the ear skin from 10-week-old mice treated for 5 weeks with anti-CSF1R or control rat IgG, stained for CD31 (red), LYVE1 (green), and counterstained with Hoechst (blue). Boxed areas are shown at higher magnification in the top right corner. Scale bar: 100 μm.

H, I    Quantification of CD31$^+$LYVE1$^-$ area (H) and number of CD31$^+$LYVE1$^-$ vessels (I).

Data information: White dotted lines in (B, D, and G) indicate the basement membrane. Statistical significance was determined using log-rank (Mantel–Cox) test (A), Mann–Whitney test (C, E, H, I) or unpaired *t*-test after log-transformation (F).

will therefore be interesting to determine the consequences of the selective loss of Tregs. Furthermore, a potential role of CD8[+] T cells in the pro-tumorigenic effect of activin should be determined, since these cells had been shown to be either pro- or anti-tumorigenic in different experimental skin cancer models (Yusuf *et al*, 2008; Kwong *et al*, 2010; Nasti *et al*, 2011).

Interestingly, we found a strong accumulation of macrophages in K14-Act and K14-Act/HPV8 mice and also in pre-tumorigenic skin of Act mice subjected to multiple TPA treatments. This is consistent with *in vitro* data showing that activin attracts human monocytes (Petraglia *et al*, 1991) and induces dendritic cell migration (Salogni *et al*, 2009). The macrophage accumulation resulted from attraction of CCR2[+] blood monocytes to the skin. This is most likely a result of the high diffusibility of activin (McDowell *et al*, 1997), resulting in increased serum levels of activin A in Act mice (Munz *et al*, 1999). Therefore, activin can induce the expression of various genes associated with cell migration/adhesion in blood monocytes. Consistent with the *in vivo* data, activin A induced monocyte/macrophage migration *in vitro* and upregulated the expression of adhesion molecules and chemokine receptors on monocyte-derived peritoneal macrophages, including *Cx3cr1,* which was shown to mediate recruitment of macrophages into several types of tumors (Held-Feindt *et al*, 2010; Schmall *et al*, 2015) and to increase macrophage survival during tumor metastasis (Zheng *et al*, 2013). Therefore, activin may not only attract monocytes through upregulation of this receptor, but also increase the survival of tissue macrophages.

F4/80 and CD11b were used for isolation of skin macrophages to obtain a broad coverage of macrophages and to allow isolation of a sufficient number of these cells from the small ear skin samples. We are aware of the limitations resulting from this sorting strategy, such as the inclusion of Langerhans cells, and it will be important in the future to further analyze the activin-induced gene expression profile in different macrophage subpopulations. Nevertheless, the data provide important novel insight into the gene expression profile of skin macrophages and also demonstrate that activin reprograms macrophages *in vivo,* resulting in a TAM-like phenotype. Thus, activin is obviously a key factor in the control of macrophage programming in the tumor microenvironment (Ruffell *et al*, 2012).

One of the activin-induced pro-tumorigenic genes encodes osteopontin, whose expression in TAMs correlated with poor prognosis in gastric cancer (Lin *et al*, 2015). Alternative translation of the *Spp1* mRNA results in intracellular or secreted variants of osteopontin (Shinohara *et al*, 2008). Intracellular osteopontin is important for migration of macrophages and innate immune receptor signaling in myeloid cells, while the secreted variant is an important cytokine in immune regulation and cancer (Inoue & Shinohara, 2011; Shevde & Samant, 2014). It promoted angiogenesis and melanoma growth (Rangaswami *et al*, 2004; Kale *et al*, 2014) as well as DMBA/TPA-induced skin carcinogenesis in mice (Hsieh *et al*, 2006). This is likely to be relevant for human skin cancer, since osteopontin is expressed in the stroma of human AK and SCCs, and promotes proliferation and survival of human keratinocytes (Luo *et al*, 2011).

Consistent with *in vitro* data (Ogawa *et al*, 2006), activin induced the expression of arginase 1 in the skin *in vivo*. This enzyme promoted tumorigenesis and keratinocyte proliferation in mice expressing a constitutively active MEK mutant in suprabasal keratinocytes, which develop tumors after skin wounding (Weber *et al*,

2016). Although arginase 1 is considered as a classical M2 marker, the activin-dependent gene expression profile was positively correlated with genes upregulated in the *in vitro* generated classically activated human macrophages. This supports the findings that activin contributes to GM-CSF-induced M1 macrophage polarization and the pro-inflammatory phenotype of these cells (Sierra-Filardi *et al*, 2011), and reflects a multipolar rather than bipolar view on macrophage activation *in vivo* (Mosser & Edwards, 2008). Thus, as in many *in vivo* models, TAMs in human SCCs showed signs of both M1 and M2 polarization (Pettersen *et al*, 2011).

Finally, activin induced the expression of several MMPs, including MMP-12, -13, and -14, resulting in increased proteolytic activity in the skin. This is likely to be functionally relevant, since increased levels of MMP-12 in myeloid cells promoted lung tumorigenesis in mice and correlated with high malignancy in human lung cancer (Hofmann *et al*, 2005; Qu *et al*, 2011). MMP-14 and its downstream effector MMP-2 are major players in angiogenesis and tumorigenesis (Kessenbrock *et al*, 2010), and MMP-13 promoted tumor angiogenesis, melanoma invasion, and metastasis (Zigrino *et al*, 2009; Kudo *et al*, 2012). The activin-induced expression of these MMPs and of additional pro-angiogenic genes in macrophages provides a likely explanation for the strong increase in the number of blood vessels in the vicinity of the epidermis of activin-overexpressing mice and thus adjacent to the source of activin.

The important role of macrophages in activin-induced skin tumorigenesis was finally demonstrated by the strong reduction in tumor formation upon macrophage depletion. This is consistent with the previously described reduction in tumorigenesis upon CSF1R inhibition in mouse models for mammary (DeNardo *et al*, 2011), cervical (Strachan *et al*, 2013), and pancreatic (Zhu *et al*, 2014) carcinoma or in glioma (Pyonteck *et al*, 2013), or upon depletion of macrophages by a DT approach or clodronate liposomes in a wound-induced skin cancer model (Arwert *et al*, 2010; Weber *et al*, 2016). Since activin expression is strongly enhanced by skin wounding (Hubner *et al*, 1996), the pro-tumorigenic effect of macrophages in this skin tumor model may be at least in part activin-dependent. In contrast, depletion of macrophages in a mouse model of BCC promoted tumor growth (Konig *et al*, 2014), suggesting that the role of macrophages is dependent on the type of tumor and most likely also on the gene expression pattern of the macrophages.

The data presented in this manuscript also provide insight into the mechanisms underlying the suppression of skin tumor development by macrophage depletion, which involves a reduction in blood vessel formation and tumor cell proliferation. These findings are consistent with the functions of TAMs in existing malignancies (Ruffell *et al*, 2012; Noy & Pollard, 2014), but also point to a crucial role of macrophage-dependent keratinocyte proliferation and angiogenesis during the early development of skin cancers. This is obviously not restricted to murine skin cancer, since our bioinformatics analysis revealed overexpression of *INHBA* in AK, together with various activin target genes in macrophages, which we had identified in our mouse model. Furthermore, expression of macrophage chemoattractant protein-1 and macrophage infiltration are associated with angiogenic promotion and poor prognosis in esophageal SCC (Koide *et al*, 2004), where activin is also overexpressed (Yoshinaga *et al*, 2003), and with malignant transformation of keratinocytes in human skin (Takahara *et al*, 2009).

Taken together, we have identified a novel role of activin in the attraction of monocytes to the skin and their reprogramming into a TAM phenotype, which is functionally important in early skin tumorigenesis. This is likely to be potentiated by its action on other cells of the tumor stroma, highlighting the strong effect of this factor on the tumor microenvironment (Antsiferova & Werner, 2012). Our results also suggest the exciting possibility of using antagonists of activin or its receptors, which are in preclinical or clinical trials for other indications (Chen *et al*, 2015; Raftopoulos *et al*, 2016), for skin cancer prevention, for example, through local treatment of skin cancer precursor lesions.

# Materials and Methods

### Human skin samples

Biopsies from normal human skin and from AK were obtained from the Department of Dermatology, University Hospital Zurich, as part of the Biobank project, approved by the local and cantonal institution review boards EK no. 647/800. Consenting patients with multiple AKs and a history of skin cancers underwent shave biopsies with a ring curette. Each sample was investigated by a board certified dermatopathologist (RD), who confirmed the diagnosis. Written informed consent was received from all patients. All research on human material abided by the Helsinki Declaration on Human Rights.

### Genetically modified mice

Act and HPV8 mice were described previously (Schaper *et al*, 2005). Act mice (in CD1 background) were crossed with HPV8 mice (in C57BL/6J or FVB1 background) to obtain double-transgenic progeny. Mice of the F1 generation of this breeding were used for the tumorigenesis experiments. Animals in tumor experiments were checked twice per week for the development of skin tumors. They were euthanized according to animal welfare regulations when a single tumor had a diameter of more than 1 cm, when more than one tumor of > 0.5 cm appeared, or when a tumor had an unfavorable localization (e.g., eye).

CD4KO and CCR2-sCFP-DTR mice (both in C57BL/6 background) and their genotyping were described before (Rahemtulla *et al*, 1991; Hohl *et al*, 2009).

Mice were housed under optimal hygiene conditions and maintained according to Swiss animal protection guidelines. All experiments with mice had been approved by the local veterinary authorities (Kantonales Veterinäramt Zürich, Switzerland).

For all animal experiments, we used mice of the same age and gender. The number of mice used for the individual experiments was based on previous experience from us and from others with similar studies (see e.g., Rolfs *et al*, 2015), taken into consideration the variability of animal studies.

### RNA isolation and qRT–PCR analysis

RNA was isolated using RNeasy Kits (Qiagen, Hilden, Germany), including a proteinase K digestion step for isolation of RNA from tissue biopsies and a DNase on-column digestion for all samples.

cDNA was synthesized using iScript Kit (Bio-Rad Laboratories, Hercules, CA, USA). Relative gene expression was determined using the Roche LightCycler 480 SYBR Green system (Roche, Rotkreuz, Switzerland) and the primers listed in Appendix Table S2.

### Histology and immunostaining

Skin and tumor biopsies were excised after the mice had been sacrificed by $CO_2$ inhalation. They were fixed in 95% ethanol/1% acetic acid for subsequent paraffin embedding or directly frozen in tissue-freezing medium (Leica Microsystems, Heerbrugg, Switzerland). Hematoxylin/eosin (H&E) and immunofluorescence staining on paraffin (7 μm) or frozen sections (4 or 10 μm) and activin A immunohistochemistry were performed as described previously (Antsiferova *et al*, 2011). For BrdU staining, mice were injected i.p. with BrdU (250 mg/kg in 0.9% NaCl; Sigma, Munich, Germany) and sacrificed 2 h after injection. Skin biopsies were stained with an antibody against BrdU. Antibodies used for immunostaining are listed Appendix Table S3. Images were acquired with Imager.A1 microscope equipped with an AxiocamMrm camera using Axiovision software (Carl Zeiss Inc., Jena, Germany) or with Pannoramic 250 Slide Scanner (3D Histech, Budapest, Hungary). The latter allowed scanning of sections covering the complete ear. Overall, the total area analyzed for each mouse ear was between 3,000 and 5,000 mm$^2$, which corresponds to approximately 30 images taken with a 20× objective.

The number of BrdU-positive, δTcR–positive, and βTcR-positive cells per mm of basement membrane, the area of CD206 positive staining, and the number of cells migrated in Transwell assay were quantified using Openlab 3.1.5 (Improvision Ltd., Basel, Switzerland) or ImageJ (NIH, Bethesda, MD, USA) software. The area, number, and size of blood vessels and the proximity of blood vessels to the basement membrane were quantified using the automated Definiens software (Definiens AG, Munich, Germany).

### Isolation of epidermal and dermal cells for magnetic-activated cell sorting and flow cytometry

Single cell suspensions of epidermal and dermal cells were prepared using modified protocols described previously (Junankar *et al*, 2006; Kisielow *et al*, 2008). Briefly, mice were sacrificed, ears were excised, separated along the cartilage, and the dorsal part was discarded. Ventral ear skin was placed on 0.2% trypsin (Invitrogen, Paisley, UK) in DMEM (Sigma) for 30 min in 37°C. After separation of the dermis, the epidermis was cut into small pieces and placed in 0.2% trypsin in DMEM supplemented with 0.25 mg/ml of deoxyribonuclease I (DNase I; Sigma) for 40 min at 37°C under continuous shaking. For flow cytometry analysis of trypsin-sensitive epitopes, ventral ear parts were placed overnight at 4°C on DMEM containing 1% penicillin/streptomycin (Sigma) and subsequently transferred to 10 mM EDTA in PBS. After 60-min incubation at 37°C, the epidermis was separated from the dermis, washed in 2 mM $CaCl_2$ in PBS, cut into pieces, and placed in DMEM supplemented with 0.25 mg/ml of DNase I for 40 min at 37°C under continuous shaking. Subsequently, DMEM supplemented with 10% FCS (Sigma) was added, the cells were filtered through 70-μm nylon strainers (BD Pharmingen, Allschwil, Switzerland), centrifuged at 300 *g* and

4°C for 10 min, and resuspended in FACS buffer (0.5% BSA, 2 mM EDTA in PBS).

To isolate cells from the dermis, the separated tissue was cut into small pieces, placed in DMEM supplemented with 2.5 mg/ml collagenase II (Worthington, Lakewood, NJ, USA), 2.5 mg/ml collagenase IV (Invitrogen, Carlsbad, CA), and 0.25 mg/ml DNase I, and incubated at 37°C for 30 min under vigorous shaking. The dermal cell suspension was filtered through a 70-μm nylon strainer, centrifuged at 300 $g$ at 4°C for 10 min, and resuspended in FACS buffer.

## MACS sorting

Magnetic cell separation sorting of epidermal and dermal cell suspensions was performed according to the recommendations of the manufacturer (Miltenyi Biotec, Bergisch Gladbach, Germany). Dead Cell Removal Kit and CD45 MicroBeads were used to purify keratinocytes from isolated epidermis.

## Flow cytometry

To prevent non-specific antibody binding, cells were incubated for 10 min at 4°C with rat anti-mouse CD16/CD32 monoclonal antibody, clone 2.4G2 (BD Pharmingen), prior to staining with epitope-specific antibodies for 30 min at 4°C. Antibodies used for flow cytometry are listed in Appendix Table S4.

To exclude non-viable cells, Zombie Aqua, Zombie Red (Biolegend, San Diego, CA, USA), 7-amino-actinomycin (7-AAD; BD Pharmingen), or Sytox Green (Invitrogen) were used according to the manufacturer's instructions. Cells were analyzed using FACS Calibur or LSRFortessa and sorted using FACS Aria IIIu equipped with the CellQuestPro or FACSDiva softwares (BD Pharmingen). Data analysis was performed using FlowJo software (Tree Star Inc., Ashland, OR, USA).

## Isolation of resident and inflammatory mouse peritoneal macrophages

Resident peritoneal macrophages were isolated from untreated mice. To obtain inflammatory thioglycollate (TG)-elicited inflammatory macrophages, mice were i.p. injected with 1 ml of 4% TG (BD Pharmingen) 4 days before isolation. They were euthanized by $CO_2$ inhalation, and the skin above the abdomen was cut and opened, leaving the muscle layer intact. 5–10 ml of cold PBS was injected into the peritoneal cavity, the abdomen was massaged to detach the cells, and the peritoneal cells were collected with a syringe and a pipet.

## Isolation of mouse blood monocytes

Blood was drawn by cardiac puncture on 5 U/ml heparin and pooled from 13 to 15 female CD1 mice. After erythrocyte lysis with ACK lysis buffer (8.29 g $NH_4Cl$, 1 g $KHCO_3$, 2 ml of 0.5 M EDTA, pH 7.3), the cells were blocked for 10 min at 4°C with rat anti-mouse CD16/CD32 monoclonal antibody (BD Pharmingen) and stained for 30 min at 4°C with 1 ml of antibody mix (Ly6G-FITC, B220-FITC, CD3-PE, CCR3-PE, CD49b-PE, MHCII-BV510, CD45-PB) to label neutrophils, B cells, T cells, eosinophils, NK cells, and dendritic cells. CD45$^+$ cells negative for the other markers

mentioned above were sorted into cold macrophage serum-free medium (Invitrogen). The purity of the sorted cells was above 88%. 70–90% of sorted cells expressed CD11b.

## Transwell migration assay

Six hundred microliter of macrophage serum-free medium (M-SFM, Invitrogen) containing recombinant human activin A (R&D Systems, Minneapolis, MN) at various concentrations was placed into the lower wells of transwell plates (Corning Incorporated, Corning, NY) with 6.5 mm diameter inserts and 8.0 μm pore size. Macrophages ($10^5$ per well, in duplicates) were seeded in 100 μl of M-SFM (with or without activin A) on the inserts. After 24 h, the inserts were removed from the plate and the upper side was cleaned with cotton swabs to remove non-migrated cells. Following fixation in cold 100% methanol for 10 min, inserts were stained with only Hoechst or together with F4/80-Alexa Fluor488 (Biolegend) for 1–3 h. After washing with PBS, membranes from inserts were cut out, placed on the glass slides, and mounted with Mowiol. The number of migrated cells was counted by blinded investigators in five 10× or 20× microscope fields and normalized to control (M-SFM).

## FACS sorting and RNA sequencing

Dorsal and ventral parts of the mouse ears were separated, minced with scissors, and digested with 0.25 mg/ml Liberase TL (Roche) in DMEM for 1 h at 37°C. Following digestion, the cell suspension was filtered through a 70-μm nylon strainer, centrifuged at 300 $g$ at 4°C for 10 min, and resuspended in FACS buffer. Ear skin macrophages were enriched by FACS of CD45$^+$F4/80$^+$CD11b$^+$ cells from the non-cancerous ear skin of 12- to 15-week-old mice. For each replicate, ear skin from 3 to 6 mice per genotype was pooled. CD3 and CD140a were used as additional negative markers. Dead cells were excluded by Sytox Green staining. The purity was above 90%, as determined by re-analysis of the sorted cells. Cells (50–200 × $10^3$ per sample) were sorted into cold empty Eppendorf tubes, centrifuged, and total RNA was isolated on the same day using the RNeasy Micro Kit (Qiagen, Hilden, Germany). RNA quality was determined on a TapeStation (Agilent, Santa Clara, CA, USA), and RIN varied from 5.1 to 6.9. RNA concentration was measured with a Qubit fluorometer (Thermo Fisher Scientific, Waltham, MA, USA), and 12 ng total RNA was used for library preparation. After ribosomal RNA depletion using the Ribo-Zero Magnetic Kit (Epicentre/Illumina, Madison, WI, USA) and a modified Protocol For Truly Low Input Total RNA Samples (Clontech, Mountain View, CA, USA), the SMARTer Stranded RNA-Seq Kit (Clontech) was used according to the manufacturer's instructions. The quality of amplified libraries was validated using a TapeStation (Agilent). Concentrations of labeled fragments were measured using absolute quantification qPCR with primers specific to Illumina sequencing adapters. Libraries were sequenced on an Illumina Hiseq 2500 v3. Approximately 20 million reads were generated per sample.

## Bioinformatic analysis of RNA-Sequencing data

Raw reads were first cleaned by removing adapter sequences, trimming low quality ends (four bases from read start and read end), and filtering reads with low quality (phred quality < 20) using

Trimmomatic (Bolger *et al*, 2014). Sequence alignment and isoform expression quantification of the resulting high-quality reads to the *Mus musculus* reference genome (build GRCm38) was performed with the RSEM algorithm (Li & Dewey, 2011) (version 1.2.18) with the option for estimation of the read start position distribution turned on. To detect differentially expressed genes, we applied the count based negative binomial model implemented in the software package edgeR (version: 3.10.2) (Robinson *et al*, 2010), in which the normalization factor was calculated by the trimmed mean of M values (TMM) method (Robinson & Oshlack, 2010). The gene-wise dispersions were estimated by conditional maximum likelihood, and an empirical Bayes procedure was used to shrink the dispersions toward a consensus value. Differential expression was assessed using an exact test adapted for over-dispersed data. Genes showing altered expression with adjusted (Benjamini and Hochberg method) *P*-value (false discovery rate, FDR) ≤ 0.05 were considered as differentially expressed. Lists of differentially expressed genes were further filtered to eliminate those genes whose normalized count was 0 in more than one sample. Venny 2.1.0 (http://bioinfogp.cnb.c sic.es/tools/venny/index.html) was used to find and visualize via a Venn diagram the common genes among the significantly differentially expressed genes. Lists of differentially expressed genes from all genotype comparisons are provided in Dataset EV2. The original data have been deposited in NCBI's Gene Expression Omnibus and are accessible through GEO Series accession number GSE79086.

### Ingenuity pathway analysis

Lists of differentially expressed genes of all three genotype comparisons (wt/Act vs. wt/wt; HPV8/Act vs. HPV8/wt; HPV8/Act vs. wt/wt) were uploaded into IPA (Qiagen), and identical core analyses using confidence of only "experimentally observed" relationships were run with filtered lists of up- and downregulated genes that exhibited an FDR value of < 0.1. Analysis-ready differentially expressed genes for each analysis were 179 for wt/Act vs. wt/wt, 207 for HPV8/Act vs. HPV8/wt, and 379 for HPV8/Act vs. wt/wt. A comparison analysis of the three core analyses was performed, and the data tables associated with "Diseases & Functions" were exported to identify the most highly activated and most highly significantly enriched functions or diseases in the three genotype comparisons. Selected functions or diseases were arranged in a spreadsheet, and their activation *z*-scores were color-coded in a pseudo-heatmap for visualization. Original IPA output data with unfiltered "Diseases & Functions" for all comparisons are provided in Dataset EV3.

### Gene set enrichment analysis

Sets of the most highly significantly upregulated (top) and downregulated (bottom) genes were generated from published microarray data. The datasets, samples, and methods for obtaining the gene sets are listed in Appendix Table S5. Original gene sets were uploaded to GSEA and filtered to those mapped by gene symbol present in the dataset being tested. The first gene sets were tested against the GSEA-generated ranked gene lists of each of the three genotype comparisons (*n* = 3; wt/Act vs. wt/wt, HPV8/Act vs. HPV8/wt, HPV8/Act vs. wt/wt) and a comparison that grouped all Act vs. wt samples (*n* = 6; [all]/Act vs. [all]/wt). Subsequent gene

sets were tested against GSEA-generated gene lists of the [all]/Act vs. [all]/wt comparison, and two microarrays of human AK vs. normal skin downloaded from the Gene Expression Omnibus (GEO): GSE2503 and GSE63107. GSEA was performed using the following settings:

| Number of permutations | 2,000 |
|---|---|
| Permutation type | Gene Set |
| Enrichment statistic | Weighted |
| Metric for ranking genes | Signal2Noise |
| Gene list sorting mode | Real |
| Normalization mode | Meandiv |
| Randomization mode | No_balance |
| Omit features with no symbol match | True |

Two separate GSEA runs were completed (i) testing all genotype comparisons against TAM gene sets and (ii) testing the [all]/Act vs. [all]/wt comparison and two human AK vs. normal skin transcriptomes against respective top upregulated gene sets, macrophage polarization gene sets, and selected myeloid/lymphoid cell-type-specific gene expression profiles. GSEA results were organized in tables, and the normalized enrichment scores were color-coded in pseudo-heatmaps for visualization. Original and filtered gene sets, ranked gene lists, and original GSEA output data for all experiments are provided in Dataset EV4.

### Preparation of protein lysates and Western blotting

Frozen tissue was homogenized in T-PER tissue protein extraction reagent (Pierce, Rockford, IL, USA) containing complete protease inhibitor cocktail and PhosSTOP phosphatase inhibitor cocktail (Roche). Lysates were cleared by sonication and centrifugation. Protein lysates were analyzed by Western blotting using osteopontin (R&D Systems) or GAPDH antibodies (HyTest, Turku, Finland). Antibody-bound proteins were detected using horseradish peroxidase coupled antibodies against goat-IgG (Calbiochem, San Diego, CA, USA) or mouse-IgG (Promega, Dübendorf, Switzerland).

### Gelatin zymography

Gelatin *in situ* and in-gel zymography were performed as previously described (Kurschat *et al*, 2002).

### Macrophage depletion *in vivo*

K14-Act/K14-HPV8 mice (CD1/FVB1 F1 background) were injected i.p. with 50 mg/kg anti-CSF1R antibody (clone AFS98, BioXCell, Upper Heyford, UK) or control rat IgG (Sigma) twice per week for 5 weeks, starting at the age of P30–P36. Only mice without obvious signs of dysplasia at the beginning of treatment were included in the experiment. Among those, randomization was performed by an investigator, who had not seen the mice, by assigning the mice to different treatment groups using the individual mouse identity number (even numbers—control IgG; odd numbers—anti-CSF1R). Signs of tumorigenic growth (scaling, hyperproliferation, ulceration) were monitored twice per week by two investigators kept blinded to

## The paper explained

### Problem

Epithelial skin cancers are the most common types of cancer in humans. Their development is frequently preceded by skin cancer precursor lesions, which need to be removed or treated due to the risk of cancer development. The incidence of these precursor lesions and of malignant skin cancers is strongly increasing due to the frequent sun exposure of the skin and the increase in aged patients, who are at particular risk for developing skin cancer and/or precursor lesions. In spite of this severe medical problem, little is known about the molecular and cellular mechanisms underlying the development of skin cancer precursor lesions and their progression into malignant skin cancers.

### Results

Here, we identify the growth and differentiation factor activin A as a novel key player in the early phase of skin cancer development. We show that activin levels are much higher in murine and human skin cancer precursor lesions compared to normal skin, and that overproduction of activin in mouse skin promotes the formation of skin tumors through recruitment of immune cells from the blood (monocytes) and their differentiation into skin macrophages with a tumor-promoting function. We isolated the macrophages from the skin of the mice prior to tumor development and analyzed their gene expression pattern. The analysis revealed that activin induces the expression of genes that promote cell migration, formation of new blood vessels and tumor cell proliferation, ultimately leading to increased tumor formation. Importantly, when we depleted the macrophages from the animals, the formation of skin tumors was strongly reduced.

### Impact

Our results provide insight into the roles and mechanisms of action of activin and macrophages in the early phase of skin tumor development. Since activin or activin receptor antagonists are in clinical trials for other disorders, this finding opens the exciting perspective of using such inhibitors for the inhibition of skin cancer development, for example, in patients with skin cancer precursor lesions.

---

the assigned treatment. Lesions persisting for more than 2 weeks were recorded as tumors.

### Statistical analysis

Statistical analysis was performed using the PRISM software, version 5.0a or 7 for Mac OS X or Windows (GraphPad Software Inc., San Diego, CA, USA). For comparison of two groups, Mann–Whitney test was performed; for comparison of more than two groups, one-way ANOVA and Bonferroni's multiple comparisons test was used; for analysis of tumor incidence, a log-rank (Mantel–Cox) test was performed. Statistical analysis of fold-change data was performed after log-transformation. For migration data, when several independent experiments were combined, each of them being normalized to corresponding control, one-sample $t$-test was used after log-transformation. For all experiments, the variance was similar between the groups that were statistically compared.

**Expanded View** for this article is available online.

### Acknowledgements

We thank Christiane Born-Berclaz for excellent technical assistance, Dr. Tamara Ramadan, Natasha Joshi, and Rodrigo Mayo (ETH Zurich) for help with mouse or tissue culture experiments, Dr. Malgorzata Kisielow and Anette Schütz from the ETH Flow Cytometry Core Facility for help with the FACS experiments, Catharine Aquino Fournier and Lennart Opitz from the Functional Genomics Center Zurich for RNA sequencing and help with the bioinformatics analysis, and Drs. Manfred Kopf and Ulrich auf dem Keller, ETH Zurich, Dr. Sigrun Smola, University of Homburg, Germany, and Dr. Cornelia Mauch, University of Cologne, Germany, for helpful suggestions. HPV8-transgenic mice and CCR2-sCFP-DTR mice were kindly provided by Dr. Herbert Pfister, University of Cologne, Germany, or Dr. Eric Pamer, Memorial Sloan Kettering Cancer Center, New York, USA, respectively. This work was supported by grants from Cancer Research Switzerland (KFS 2822-08-2011 and KFS-3474-08-2014 to S.W.), the Wilhelm Sander-Stiftung (to S.W.), the Swiss National Science Foundation (310030_132884 to S.W.), and a Marie Curie postdoctoral fellowship from the European Commission (to A.P-C).

### Author contributions

MA performed experiments, analyzed and interpreted data, designed and planned the study together with SW, and wrote the manuscript together with SW. AP-C, MC, MW, DS, and KB performed experiments and analyzed data, DH performed histopathological analysis of pre-tumorigenic mouse skin and mouse tumors, RD, ML, and VCA provided and prepared human AK samples, RD performed their histopathological analysis, and SW designed and planned the study together with MA, wrote the manuscript together with MA, and provided the funding.

### Conflict of interest

The authors declare that they have no conflict of interest.

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
