## [Review Process File · EMBO Molecular Medicine]

Activin promotes skin carcinogenesis by attraction and reprogramming of macrophages

Maria Antsiferova, Aleksandra Piwko-Czuchra, Michael Cangkrama, Mateusz Wietecha, Dilara Sahin, Katharina Birkner, Valerie C. Amann, Mitchell Levesque, Daniel Hohl, Reinhard Dummer, and Sabine Werner

Corresponding author: Sabine Werner, ETH Zurich

Review timeline:

Submission date:	17 April 2016
Editorial Decision:	13 June 2016
Revision received:	04 September 2016
Editorial Decision:	11 October 2016
Revision received:	25 October 2016
Accepted:	28 October 2016

Transaction Report:

Editor: Roberto Buccione

1st Editorial Decision

13 June 2016

Thank you for the submission of your manuscript to EMBO Molecular Medicine. We are sorry that it has taken much longer than usual to get back to you on your manuscript. In this case we experienced difficulties in securing three appropriate expert reviewers, and then obtaining their evaluations in a timely manner.

As you will see the Reviewer 1 is quite positive, Reviewers 2 and 3 are much more reserved and raise several issues of consequence on different aspects, some of which overlapping, including the unclear role of macrophage recruitment in activin-induced carcinogenesis, the potential role of T cell populations other than CD4+, experimental issues on macrophage re-programming, and insufficient human data.

There are also concerns on novelty. While for instance, #2 feels that novelty is significant with respect to the HPV8/activin cooperation in promoting skin cancer and the macrophage recruiting action of activin, #3 is much less convinced of the news value with reference to the known role of activin in skin carcinogenesis. I feel that they are both right and the onus is on you to clarify these aspects.

I also note that Reviewers #1 and 3 clearly feel that data presentation and image quality has room for much improvement.

Reviewer 2 also suggests that the manuscript should be split to tell two separate stories. I do not necessarily agree, but I do suggest that an effort should be made to better harmonise the study.

In conclusion, while publication of the paper cannot be considered at this stage, given the potential interest of your findings, we have decided to give you the opportunity to address the above concerns. We are thus prepared to consider a substantially revised submission, with the understanding that the Reviewers' concerns must be addressed with additional experimentation as appropriate and that acceptance of the manuscript will entail a second round of review.

Please note that it is EMBO Molecular Medicine policy to allow a single round of revision only and that, therefore, acceptance or rejection of the manuscript will depend on the completeness of your responses included in the next, final version of the manuscript.

As you know, EMBO Molecular Medicine has a "scooping protection" policy, whereby similar findings that are published by others during review or revision are not a criterion for rejection. However, I do ask you to get in touch with us after three months if you have not completed your revision, to update us on the status. Please also contact us as soon as possible if similar work is published elsewhere.

As mentioned above, EMBO Molecular Medicine now requires a complete author checklist (<http://embomolmed.embopress.org/authorguide#editorial3>) to be submitted with all revised manuscripts. Provision of the author checklist is mandatory at revision stage; The checklist is designed to enhance and standardize reporting of key information in research papers and to support reanalysis and repetition of experiments by the community. The list covers key information for figure panels and captions and focuses on statistics, the reporting of reagents, animal models and human subject-derived data, as well as guidance to optimise data accessibility.

We now mandate that all corresponding authors list an ORCID digital identifier. You may do so though our web platform upon submission and the procedure takes < 90 seconds to complete. We also encourage co-authors to supply an ORCID identifier, which will be linked to their name for unambiguous name identification.

Please carefully adhere to our guidelines for authors (<http://embomolmed.embopress.org/authorguide>) to accelerate manuscript processing in case of acceptance.

I look forward to seeing a revised form of your manuscript as soon as possible.

***** Reviewer's comments *****

Referee #1 (Comments on Novelty/Model System):

The authors present a detailed analysis of their data with logical progression of ideas. Many people would not regard mouse-HPV induced papillomas as comparable to human SCC. The authors are careful not to conclude that the model is a SCC model, they are looking at a skin carcinogenesis model.

Referee #1 (Remarks):

This is an interesting paper demonstrating that activin induces skin carcinogenesis via attraction and reprogramming of macrophages and also identifies novel activin targets involved in tumor formation. The scientific and technical value of the work is high. I have the following comments.

The authors show that about one third of the AKs examined have increased activin expression. In fact, far less than one third of AKs progress to SCC. It may be that a different genetic subtype of AK has activin overexpression.

In the mouse HPV papilloma model, a lot of the papillomas have trichoepitheliomatous differentiation, suggesting that they are derived from the hair follicle. Although activin overexpression increases the number of papillomas it does not seem to have an influence on the number of trichoepitheliomas or papillomas with trichoepitheliomatous differentiation.

Some further detail re stats would be useful eg a comparison of n=2 wt/Act mice versus n= 8 HPV8/Act mice was significant (Figure 3B)? Were there multiple sections/images analysed etc?

Some of the pictures are pixellated and I think the Meca32 staining could be improved. This usually works very well in mouse sections.

In the supplementary data spreadsheets in the comparison of Act HPV and HPV mice the most upregulated gene was Krt17 which is expressed in the hair follicle, which would go along with the notion of Activin driving hair follicle tumourigenesis.

Despite these reservations, I think the data presented are very interesting and provide a paradigm for novel therapy of pre-skin cancer.

Some

Referee #2 (Comments on Novelty/Model System):

Overall, this is an interesting but complex study. The novelty is two-fold. First, the potential cooperation between HPV8 and activin to promote skin cancers. Second, the role of activin in recruiting macrophages is also novel. However, two major weaknesses decrease the enthusiasm for the study. First, one would like to know how HPV8 and activin cooperate to promote tumorigenesis. Second, it is unclear how macrophage recruitment by activin relates to HPV8.

One wonder whether the investigators may instead prepare two distinct manuscripts focusing on the two main findings and add mechanistic studies to increase the significance of their findings.

Referee #2 (Remarks):

In this manuscript, the investigators have explored the role of activin together with human papilloma virus 8 (HPV8) to promote skin tumorigenesis. They have already reported the role of activin in promoting skin tumorigenesis by inducing Langerhans and regulatory T cell infiltration and inhibition the proliferation of gamma delta T cells.

Here, using genetically-engineered mice expressing HPV8 oncogenes with or without activin in keratinocytes, the investigators observe that HPV8 oncogenes and activin cooperate in skin cells to promote skin tumors. They observe the loss of gamma delta cells and presence of alpha beta T cells. Skin tumorigenesis does appear to depend on CD4+ T cells. Finally, they report increased the numbers of macrophages in the skin, which exhibit a pro-tumoral phenotype

Overall, this is an interesting but complex study. The novelty is two-fold. First, the potential cooperation between HPV8 and activin to promote skin cancers. Second, the role of activin in recruiting macrophages is also novel. However, two major weaknesses decrease the enthusiasm for the study. First, one would like to know how HPV8 and activin cooperate to promote tumorigenesis. Second, it is unclear how macrophage recruitment by activin relates to HPV8.

Additional comments

1. The investigators, report T cell infiltration, based on 5 cases. One would like to see more cases as well as representative flow cytometry data.
2. The investigators have performed CD4 depletion but not CD8 T cell depletion. Therefore, one cannot rule out that CD8+ T cells may play a role in reducing the tumor-promoting effect of activin.
3. The data on macrophages are interesting. However, it is unclear how these relate to HPV8, and additional studies are needed to support the role of the macrophage in activin-induced skin tumorigenesis.

Referee #3 (Comments on Novelty/Model System):

This is mainly a border-line paper regarding novelty, as detailed in my review.

Referee #3 (Remarks):

The manuscript by Antsiferova et al. entitled "Activin promotes skin carcinogenesis by attraction and re-programming of macrophages" confirms the already known role of Activin in skin carcinogenesis, as previously described by the same group (Antsiferova et al. *Nature Commun.*, 2011). Compared to the previous article, the authors provide an additional mechanism by which Activin promotes tumorigenesis in a clinically more relevant skin cancer model induced by Human Papilloma Virus 8 (HPV8), through the recruitment and re-programming of macrophages. The authors combined keratinocyte-specific expression of HPV8 oncogenes and Activin under the control of the keratin-14 promoter. The result is a dramatic increase in tumor incidence with similar features regarding T cells, as observed in the previous publication using a chemically-induced skin carcinogenesis mouse model. Along with these observations, a significant increase in macrophages are mobilized to the skin in response to Activin. Skin macrophages were characterized by large-scale transcriptomic analysis to resemble tumor-associated macrophages (TAMs), suggesting a pro-tumorigenic role of these cells. Global depletion of macrophages delayed tumor development in the Activin-dependent HPV8-induced skin cancer model.

Although there is not much novelty regarding the role of Activin in cutaneous SCCs, the effect of epithelial-secreted Activin on macrophages is interesting that might be relevant to other diseases, where Activin is functionally involved, and warrants further investigations. However, there are some inconsistencies in the way the data are interpreted and the manuscript should be subjected to major revisions before being considered for publication in *EMBO Molecular Medicine*.

Major points:

1. The paper focuses on a pre-tumorigenic stage of the skin cancer model that varies from 10-16 weeks of age depending on the experiment. For instance, the macrophage isolation for the transcriptomic analysis was done with 13-15 week-old mice from all groups. According to Figure 1F, by 15 weeks of age more than 50% of HPV8/Act mice have visible tumors. This means that macrophages isolated from 13-15 week-old mouse skin cannot be considered pre-tumorigenic. Actually, subsequent analysis should take into account the probable presence of macrophages associated to a growing tumor that could be TAMs. This is a very important point as it is the only novel aspect of this manuscript. The authors should re-word their conclusions about the pre-tumorigenic macrophages, and carry out additional experiments at an earlier time point, where there is no tumor in either group. Particularly, the experiments showing a re-programming of macrophages should be validated at an earlier stage.
2. The control HPV8/wt group is missing in the anti-Csf-1r therapy experiment and therefore it is difficult to know whether the result is due to the effect only on Activin actions, or it would also be effective in a model without forced Activin expression. Therapy could also affect directly some of the measured parameters, such as keratinocyte proliferation. Additionally, it would be more appropriate to show macrophage depletion through a quantitative method, such as flow cytometry.
3. The human data need to be expanded. It would be desirable to see some validation of the correlation of cutaneous SCC/Activin/macrophages in human samples.
4. Figure 1B shows that the skin tumors of HPV8/wt mice have endogenous upregulation of activin. However, this upregulation does not translate into increased recruitment or reprogramming of macrophages compared to wt mice. This point should be more clearly addressed/discussed as the major conclusion of the paper is the fact that Activin expression is responsible for this recruitment/reprogramming of macrophages in skin tumors.
5. Macrophage markers used in the different techniques are not consistent: MerTK/CD64 for flow cytometry analysis in Figure 3, CCR2 expressing cells for macrophage depletion by diphtheria toxin and finally F4-80/CD11b for cell sorting and gene expression analysis. Macrophages are a very

heterogeneous population and it is likely that these strategies identify different subtypes. This should be made more consistent and taken into account for the interpretation of the data. The diversity of markers used in different experiment has to be discussed as a limitation of the study. Furthermore, it would be of interest to differentiate between resident vs. recruited macrophages.

6. The authors previously showed the effects of Activin on skin tumorigenesis using the DMBA/TPA protocol. It would be important to determine whether the macrophage mechanisms described in this manuscript also hold true in the DMBA/TPA carcinogenesis model.

Minor points:

1. The immunofluorescence images throughout the paper are very hard to interpret. Most of them need to be replaced by higher magnification images.
2. Ingenuity Pathway Analysis and Gene Set Enrichment Analysis in Figure 4 are busy and hard to interpret. It would be advisable to depict the data in a manner easier for the reader to appreciate.

1st Revision - authors' response

04 September 2016

Referee #1 (Remarks):

This is an interesting paper demonstrating that activin induces skin carcinogenesis via attraction and reprogramming of macrophages and also identifies novel activin targets involved in tumor formation. The scientific and technical value of the work is high. I have the following comments.

The authors show that about one third of the AKs examined have increased activin expression. In fact, far less than one third of AKs progress to SCC. It may be that a different genetic subtype of AK has activin overexpression.

Our reply: This point is well taken and we have therefore discussed this in more detail (page 20). We also cite a recent manuscript demonstrating that 2.57% of AKs progress to SCC during a 4-year interval (Criscione et al., 2009). Activin overexpression is certainly not the only risk factor for malignant progression, but it could contribute to it based on our previous findings in the DMBA/TPA skin carcinogenesis model.

In the mouse HPV papilloma model, a lot of the papillomas have trichoepitheliomatous differentiation, suggesting that they are derived from the hair follicle. Although activin overexpression increases the number of papillomas it does not seem to have an influence on the number of trichoepitheliomas or papillomas with trichoepitheliomatous differentiation.

Our reply: Fig. EV1G shows that 9 acanthopapillomas with trichoepitheliomatous differentiation were detected in HPV8/wt mice, but 43 of such tumors were seen in HPV8/Act mice. Therefore, there is also a strong increase in the number of these tumors in the presence of the activin transgene. In addition, 6 tumors in HPV8/Act mice were trichoepitheliomas, but none in HPV8/wt mice. We now clarify this issue in the text (Results, page 8).

Some further detail re stats would be useful eg a comparison of n=2 wt/Act mice versus n= 8 HPV8/Act mice was significant (Figure 3B)? Were there multiple sections/images analysed etc?

Our reply: Only the differences indicated in the figures with an asterisk are statistically significant. The difference between wt/Act and HPV8/Act (Fig. 3b) was not statistically significant. The low number of wt/Act mice that we had in this experiment due to the unfortunate distribution of the genotypes does not allow a statistical analysis. However, we verified the results by flow cytometry using more mice per genotype (and different animals) in several independent experiments (e.g. Fig. 3C, D, new version). Therefore, it was reproducibly observed that activin strongly increases the number of macrophages, whereas the HPV8 transgene has no or only a very minor effect on the number of macrophages.

We now clarify that for the analysis of stained area we scanned 1 section using a 3D Histotech Slide Scanner, which covered the complete ear. Overall, the total area analyzed for each mouse was between 3000 and 5000 mm², which corresponds to approximately 30 images taken at 20x

objective. This is now mentioned in Materials and Methods.

Some of the pictures are pixellated and I think the Meca32 staining could be improved. This usually works very well in mouse sections.

Our reply: We looked at the high magnification pictures and the quality is generally very good. The pixelated appearance obviously results from the low resolution PDF that was made during the conversion. We have now uploaded the original figures. In addition, we now show higher magnifications of the Meca32 staining and of some other stainings (see also request of reviewer 2). The old figures have been replaced by improved figures (Fig. 5 and 6).

In the supplementary data spreadsheets in the comparison of Act HPV and HPV mice the most upregulated gene was Krt17 which is expressed in the hair follicle, which would go along with the notion of Activin driving hair follicle tumourigenesis.

Our reply: Although the macrophage population that we isolated was more than 95% pure as mentioned in Materials and Methods, some minor contamination with other cells of the skin cannot be avoided. *K17* is a gene that is specifically expressed in hair follicle keratinocytes and therefore, detection of keratin 17 mRNA reflects a minor contamination with hair follicle keratinocytes. This contamination occurred in 11 out of 12 samples. In one sample *K17* was not detectable and in the other samples the expression was very low (average of 60 in normalized counts, as compared to 560 for *Cx3cr1*, for example). Also, upon closer inspection we realized that some of the genes in the original lists, including *K17*, were not $FDR < 0.05$ and $Log_2Ratio > 1$. Since this is misleading, we should have removed all the genes from the lists, which do not strictly satisfy the statistical criteria indicated and which have 0 expression more than one sample being compared. This has now been done, and the change affects *K17*, *Tlr12*, *Orc1* and about a dozen others which appeared in the original lists, but none of the focus genes that are robustly expressed in macrophages and important for their function. The additional filtering slightly affected the Venn diagram and heatmap in Fig 4F,G and thus these figures have been updated in the revised version.

Despite these reservations, I think the data presented are very interesting and provide a paradigm for novel therapy of pre-skin cancer.

Referee #2 (Comments on Novelty/Model System):

Overall, this is an interesting but complex study. The novelty is two-fold. First, the potential cooperation between HPV8 and activin to promote skin cancers. Second, the role of activin in recruiting macrophages is also novel. However, two major weaknesses decrease the enthusiasm for the study. First, one would like to know how HPV8 and activin cooperate to promote tumorigenesis. Second, it is unclear how macrophage recruitment by activin relates to HPV8.

One wonder whether the investigators may instead prepare two distinct manuscripts focusing on the two main findings and add mechanistic studies to increase the significance of their findings.

Our reply: We have carefully considered the suggestion of the reviewer to split the manuscript. However, the editor was not enthusiastic about this possibility and we also believe that the data belong together to make a more complete story. We have, however, added additional data and explanations to improve the manuscript.

Referee #2 (Remarks):

In this manuscript, the investigators have explored the role of activin together with human papilloma virus 8 (HPV8) to promote skin tumorigenesis. They have already reported the role of activin in promoting skin tumorigenesis by inducing Langerhans and regulatory T cell infiltration and inhibition the proliferation of gamma delta T cells.

Here, using genetically-engineered mice expressing HPV8 oncogenes with or without activin in keratinocytes, the investigators observe that HPV8 oncogenes and activin cooperate in skin cells to promote skin tumors. They observe the loss of gamma delta cells and presence of alpha beta T cells. Skin tumorigenesis does appear to depend on CD4+ T cells. Finally, they report increased the numbers of macrophages in the skin, which exhibit a pro-tumoral phenotype

Overall, this is an interesting but complex study. The novelty is two-fold. First, the potential cooperation between HPV8 and activin to promote skin cancers. Second, the role of activin in recruiting macrophages is also novel. However, two major weaknesses decrease the enthusiasm for the study. First, one would like to know how HPV8 and activin cooperate to promote tumorigenesis. Second, it is unclear how macrophage recruitment by activin relates to HPV8.

Our reply:

Cooperation between HPV8 and activin to promote tumorigenesis:

Development of epithelial skin cancers requires an oncogenic stimulus in keratinocytes. In the case of the HPV8 mice the oncogenic stimulus is provided by the expression of the oncogenes of HPV8 in keratinocytes. In the DMBA/TPA model it results from DMBA-induced mutations in keratinocytes, in particular in the *ras* proto-oncogene. Activin alone does not provide an oncogenic stimulus and we have never seen any tumor in mice overexpressing only activin (and not the HPV8 oncogenes). We have clarified this in the text (page 7). In fact, our previous experiments in the DMBA/TPA model demonstrated that activin exerts even a mild tumor-suppressive effect via keratinocytes, but this is overruled by the strong pro-tumorigenic effect that is entirely mediated via the stroma. We realized that we had not provided enough information on these data in the Introduction and we have therefore included further information in the revised version. Therefore, the mechanism of the cooperation is obvious: HPV8 is required for malignant transformation of keratinocytes, and tumor development/growth is accelerated by activin through its effect on the stroma. In a search for the relevant stromal cells we now identified macrophages as key players. This effect is most likely potentiated by additional pro-tumorigenic effects of activin on other stromal cells. We have further clarified this issue in the discussion (page 26).

Macrophage recruitment by activin and its relation to HPV8:

As shown in Fig. 3B and C, the recruitment of macrophages occurred already in wt/Act mice and thus did not require the HPV8 transgene. In fact, the HPV8 transgene had no or only a minor effect on the number of macrophages (Fig. 3B,C) and on the macrophage gene expression pattern (Dataset EV2). We now mention this explicitly in the text. In addition, we now show data from the DMBA/TPA carcinogenesis study demonstrating that activin also attracts macrophages under these conditions. These new results are now shown in Fig. 3D. Therefore, macrophage recruitment by activin does not require the HPV8 transgene.

Additional comments

1. The investigators, report T cell infiltration, based on 5 cases. One would like to see more cases as well as representative flow cytometry data.

Our reply: As requested by the reviewer, we now analyzed the $\alpha\beta$ T cells in more mice (8 mice in total). This did not change the result and the difference between genotypes is now even more significant. The figure has been replaced by a new figure, which includes the additional mice. We also show the representative flow cytometry data in Appendix Fig. 1. We would also like to point out that we verified the data by flow cytometry for β TcR and either CD4 or CD8 using independent animals. In this case we used pools from 3 mice in each experiment and the experiment was repeated 5 times as mentioned in the Legend to Fig. 2. Therefore, 15 mice per genotype were used for this study.

2. The investigators have performed CD4 depletion but not CD8 T cell depletion. Therefore, one cannot rule out that CD8+ T cells may play a role in reducing the tumor-promoting effect of activin.

Our reply: We decided to focus on CD4 T cells, since CD4 T cells promote malignant progression of skin tumors in mice expressing an HPV16 transgene in keratinocytes (Daniel et al., 2003) and tumor formation in the DMBA/TPA- and UV-induced skin cancer models (Yusuf et al., 2011; Nasti et al., 2011). By contrast, CD8 T cells were tumor-protective in DMBA/TPA-induced and UV-induced skin cancer models (Yusuf et al., 2008 and 2011). In addition, we observed a strong accumulation of CD4⁺ regulatory T cells and therefore, analysis of the consequences of the loss of this population was the focus of our study. However, tumor-promotive effects of CD8⁺ T cells have indeed also been observed in skin carcinogenesis studies (see for example Kwong et al., 2010) and therefore, we now discuss this possibility in our manuscript (page 22). This is certainly an

interesting experiment for future studies, but will require generation and analysis of triple mutant mice, which takes approximately two years as we know from our CD4 T cell study. Since our manuscript focuses on macrophages, we believe that another T cell study would extend the scope of our manuscript.

3. The data on macrophages are interesting. However, it is unclear how these relate to HPV8, and additional studies are needed to support the role of the macrophage in activin-induced skin tumorigenesis.

Our reply: As mentioned in our response to the general comments of this reviewer, we now clarify that attraction of macrophages by activin does not require the *HPV8* transgene. In particular, we now show data from the DMBA/TPA carcinogenesis study demonstrating that activin also attracts macrophages under these conditions. These new results are now shown in Fig. 3D. The DMBA/TPA data further support the role of macrophages in activin-induced tumorigenesis. Finally, we have done a thorough bioinformatics analysis using published RNA profiling data from human AK versus normal skin and from *in vitro* differentiated human macrophages. The analysis confirmed the upregulation of *INHBA* mRNA in AK. Most importantly, this upregulation correlated with upregulation of major activin target genes in macrophages identified in our study, including *ARG1*, *STAT1*, *F11R* and others. Furthermore, a comparison of the AK data with RNA profiling data from *in vitro* differentiated human macrophages demonstrated upregulation of genes expressed upon macrophage differentiation in human AK. These results are now shown in Fig. 4E and described in the text.

Referee #3 (Comments on Novelty/Model System):

This is mainly a border-line paper regarding novelty, as detailed in my review.

Referee #3 (Remarks):

The manuscript by Antsiferova et al. entitled "Activin promotes skin carcinogenesis by attraction and re-programming of macrophages" confirms the already known role of Activin in skin carcinogenesis, as previously described by the same group (Antsiferova et al. Nature Commun., 2011). Compared to the previous article, the authors provide an additional mechanism by which Activin promotes tumorigenesis in a clinically more relevant skin cancer model induced by Human Papilloma Virus 8 (HPV8), through the recruitment and re-programming of macrophages. The authors combined keratinocyte-specific expression of HPV8 oncogenes and Activin under the control of the keratin-14 promoter. The result is a dramatic increase in tumor incidence with similar features regarding T cells, as observed in the previous publication using a chemically-induced skin carcinogenesis mouse model. Along with these observations, a significant increase in macrophages are mobilized to the skin in response to Activin.

Skin macrophages were characterized by large-scale transcriptomic analysis to resemble tumor-associated macrophages (TAMs), suggesting a pro-tumorigenic role of these cells. Global depletion of macrophages delayed tumor development in the Activin-dependent HPV8-induced skin cancer model.

Although there is not much novelty regarding the role of Activin in cutaneous SCCs, the effect of epithelial-secreted Activin on macrophages is interesting that might be relevant to other diseases, where Activin is functionally involved, and warrants further investigations. However, there are some inconsistencies in the way the data are interpreted and the manuscript should be subjected to major revisions before being considered for publication in EMBO Molecular Medicine.

Our reply: We agree with the reviewer that we had already shown a pro-tumorigenic effect of activin in the skin in our previous paper. However, a lot of open questions obviously remain after a first publication of a novel role of a certain gene in cancer. In this respect we would like to mention the multitude of articles that were published in high impact journals on the role of TGF- β in cancer, including skin cancer. Our new study includes several major novel findings:

- a.) Insight into the cellular and molecular mechanisms of activin action in the skin (effect on the number, differentiation and gene expression pattern of macrophages, functional data on the role of macrophages in the pro-tumorigenic effect of activin)
- b.) Identification of a role of activin in the early phase of skin carcinogenesis

- c.) First transcriptomic analysis of macrophages in pre-tumorigenic skin
- d.) Role of activin in a physiologically more relevant skin cancer model

In fact, reviewer 2 explicitly mentions that these findings are novel, in particular the effect of activin on macrophages *in vivo*.

Major points:

1. *The paper focuses on a pre-tumorigenic stage of the skin cancer model that varies from 10-16 weeks of age depending on the experiment. For instance, the macrophage isolation for the transcriptomic analysis was done with 13-15 week-old mice from all groups. According to Figure 1F, by 15 weeks of age more than 50% of HPV8/Act mice have visible tumors. This means that macrophages isolated from 13-15 week-old mouse skin cannot be considered pre-tumorigenic. Actually, subsequent analysis should take into account the probable presence of macrophages associated to a growing tumor that could be TAMs. This is a very important point as it is the only novel aspect of this manuscript. The authors should re-word their conclusions about the pre-tumorigenic macrophages, and carry out additional experiments at an earlier time point, where there is no tumor in either group. Particularly, the experiments showing a re-programming of macrophages should be validated at an earlier stage.*

Our reply: This is indeed an important point and we apologize if this was not sufficiently clarified in the initial version of our manuscript. We performed several skin tumorigenesis experiments, which reproducibly showed the strong pro-tumorigenic effect of activin. The first experiments, for which the results are shown in Fig. 1 and Fig. 2, had been performed in our previous mouse facility. During the course of the study we had to move to another facility, which required embryo transfer. Although both facilities were/are pathogen-free, the hygiene standard in the new facility is even higher and the environmental conditions are slightly different. After the move to the new facility, we still observed the same strong pro-tumorigenic effect of activin and its effect on macrophages, but the appearance of the tumors was delayed in the HPV8/Act and HPV8/wt mice and even more in the HPV8/wt mice compared to the situation in the old facility. Therefore, the macrophage isolation was performed in the new facility with mice at the age of 13-15 weeks. None of these mice had any tumor at any body site – we checked this carefully for each individual mouse. Therefore, the macrophages were not associated with a growing tumor. We have now clarified this important issue in the next (Results, page 12), and the tumor incidence in both facilities is shown below for the reviewer (left panel: old facility; right panel: new facility)

Furthermore, the expression profile of macrophages from Act/HPV8 mice is almost identical to that of Act single transgenic animals, pointing out that overexpression of activin is sufficient to induce a pro-tumorigenic gene expression signature in skin macrophages.

2. *The control HVP8/wt group is missing in the anti-Csf-1r therapy experiment and therefore it is difficult to know whether the result is due to the effect only on Activin actions, or it would also be effective in a model without forced Activin expression. Therapy could also affect directly some of the measured parameters, such as keratinocyte proliferation. Additionally, it would be more appropriate to show macrophage depletion through a quantitative method, such as flow cytometry.*

Our reply: We agree that it would be ideal to have this control included in the experiment. However, spontaneous tumor development in HPV8/wt mice occurs very late, and in our new facility it occurs even later. Therefore, we would have to inject the mice for several months and the mice would

suffer too much from two injections per week during such a long period. We would not even get permission from the veterinary authorities to do such an experiment. Treatment of HPV8/wt mice at the same age as HPV8/Act mice and for the same period of time would thus not allow us to observe any effect on tumor formation. In addition, any effect of infiltrated/accumulated macrophages on other parameters could not be studied, since there is no macrophage infiltration in the HPV8/Act at this early age. Therefore, we have decided not to perform this experiment to avoid use of additional animals, and we hope that the reviewer agrees with this decision.

The material that we obtained from ear skin was very limited and we therefore had to decide carefully for what purpose to use it. Flow cytometry would have required a significant part of the ear skin and we therefore decided to use the material for histology/immunostaining as well as for RNA isolation. Thus, macrophage depletion was verified by immunohistochemistry and quantification of the CD68 positive area. We now further validated the depletion of macrophages by immunostaining for CD206, and these new data are now shown in Fig. 6D,E. We would also like to mention that MMP12 is predominantly expressed in macrophages. Therefore, the strong reduction in the expression of this gene after macrophage depletion provides additional quantitative data for the loss of these cells. This is now mentioned in the text (page 18/19).

3. The human data need to be expanded. It would be desirable to see some validation of the correlation of cutaneous SCC/Activin/macrophages in human samples.

To address this important comment of the reviewer, we performed a thorough bioinformatics analysis using published RNA profiling data from human AK versus normal skin and from *in vitro* differentiated macrophages. The analysis confirmed the upregulation of *INHBA* mRNA in AK. Most importantly, this upregulation correlated with upregulation of major activin target genes in human macrophages identified in our study. Furthermore, a comparison of the AK data with RNA profiling data from *in vitro* differentiated human macrophages demonstrated upregulation of genes expressed upon macrophage differentiation in human AK. These results are now shown in Fig. 4E and described in the text.

4. Figure 1B shows that the skin tumors of HPV8/wt mice have endogenous upregulation of activin. However, this upregulation does not translate into increased recruitment or reprogramming of macrophages compared to wt mice. This point should be more clearly addressed/discussed as the major conclusion of the paper is the fact that Activin expression is responsible for this recruitment/reprogramming of macrophages in skin tumors.

Our reply: The increase in endogenous activin expression in the pre-tumorigenic skin of HPV8 mice compared to skin of wild-type mice is very mild and non-significant (Fig. 1B). Therefore, it is not surprising that there is no increase in the number of macrophages in the HPV8/wt mice at this stage. We have clarified this important point in the text (page 6). Rather, the increase only occurs in the benign tumors of HPV8 mice.

5. Macrophage markers used in the different techniques are not consistent: MerTK/CD64 for flow cytometry analysis in Figure 3, CCR2 expressing cells for macrophage depletion by diphtheria toxin and finally F4-80/CD11b for cell sorting and gene expression analysis. Macrophages are a very heterogeneous population and it is likely that these strategies identify different subtypes. This should be made more consistent and taken into account for the interpretation of the data. The diversity of markers used in different experiments has to be discussed as a limitation of the study. Furthermore, it would be of interest to differentiate between resident vs. recruited macrophages.

Our reply: The heterogeneity of macrophages is certainly a very important issue. We initially observed the accumulation of macrophages in the skin of Act mice upon F4/80 immunostaining and F4/80⁺/CD11b⁺ flow cytometry. Since F4/80 and CD11b are not exclusively expressed on macrophages, we verified this finding using flow cytometry with antibodies against MerTK and CD64 as well as by immunostaining for CD206. We used the CCR2-eCFP-DTR mice for the depletion experiment, since CCR2 is a marker for monocytes. The CCR2 protein is required for emigration of these cells from the bone marrow (Serbina and Pamer, *Nat. Immunol.*, 2006). This strategy therefore allowed us to determine if the macrophages that accumulate in the skin of Act mice are derived from the bone marrow. We have now clarified this further in the text. In this experiment, detection of skin macrophages was also performed by flow cytometry with antibodies for CD64 and MerTK, demonstrating that CD64⁺/MerTK⁺ macrophages are lost in the skin upon

depletion of CCR2-positive blood-borne precursors. This experiment also differentiates between resident and recruited macrophages and thus answers this question of the reviewer.

We agree that we could have used MerTK and CD64 for the sorting. However, we decided to use a broader myeloid marker for the sorting to determine global effects of activin on the myeloid cell population. This approach was also necessary to obtain enough sorted cells from ear skin, which was particularly difficult in the mice lacking the activin transgene. We have now clarified this issue in the text and we mention this potential limitation as requested by the reviewer (page 22/23).

6. The authors previously showed the effects of Activin on skin tumorigenesis using the DMBA/TPA protocol. It would be important to determine whether the macrophage mechanisms described in this manuscript also hold true in the DMBA/TPA carcinogenesis model.

Our reply: As requested by the reviewer, we now show data from the DMBA/TPA carcinogenesis study, demonstrating that activin overexpression also correlates with an increase in skin macrophages under these conditions. These new results are now shown in Fig. 3D. They further support the role of macrophages in activin-induced tumorigenesis.

Minor points:

1. The immunofluorescence images throughout the paper are very hard to interpret. Most of them need to be replaced by higher magnification images.

Our reply: As requested by the reviewers, we have added higher magnification images for most of the immunofluorescence images. We would like to point out, however, that the rather poor quality is a problem with the low resolution PDF that we had to submit. We now submit the original files at high magnification and we hope that the PDF sent to the reviewers will be of sufficient quality.

2. Ingenuity Pathway Analysis and Gene Set Enrichment Analysis in Figure 4 are busy and hard to interpret. It would be advisable to depict the data in a manner easier for the reader to appreciate.

Our reply: As requested by the reviewer, we now show a less busy presentation of the data (Fig. 4C,D,E, new version), combined with a more detailed description in the legends.

2nd Editorial Decision

11 October 2016

Thank you for the submission of your revised manuscript to EMBO Molecular Medicine. We have now received the enclosed reports from the Reviewers who were asked to re-assess it.

As you will see, while reviewer 2 is now satisfied with the manuscript, reviewer 3 is still concerned that the news value of this manuscript is too limited. Reviewer 3 has also a few remaining concerns connected to your rebuttal.

While I will not be considering the novelty issue at this stage of development also because we agree with the other two reviewers who instead recognized the various new findings, I must ask you to carefully address the remaining concerns, including point 1 on the impact of the new mouse facilities. I am prepared to evaluate your next, final version of your manuscript at the editorial level, provided the concerns are fully addressed.

Please also comply with the following editorial requests:

- 1) Please provide a higher quality version of Figure 5J.
- 2) As per our Author Guidelines, the description of all reported data that includes statistical testing must state the name of the statistical test used to generate error bars and P values, the number (n) of independent experiments underlying each data point (not replicate measures of one sample), and the actual P value for each test (not merely 'significant' or ' $P < 0.05$ ').
- 3) We are now encouraging the publication of source data, particularly for electrophoretic gels and blots, with the aim of making primary data more accessible and transparent to the reader. Would you be willing to provide a PDF file per figure that contains the original, uncropped and unprocessed scans of all or at least the key gels used in the manuscript? The PDF files should be labeled with the

appropriate figure/panel number, and should have molecular weight markers; further annotation may be useful but is not essential. The PDF files will be published online with the article as supplementary "Source Data" files. If you have any questions regarding this just contact me.

Please submit your revised manuscript within two weeks. I look forward to seeing a revised form of your manuscript as soon as possible.

***** Reviewer's comments *****

Referee #2 (Comments on Novelty/Model System):

The authors have answered to most part of the the main concerns raised at the previous review.

Referee #3 (Comments on Novelty/Model System):

See previous review and still border-line case re novelty.

Referee #3 (Remarks):

Unfortunately, only some comments have been appropriately addressed as detailed below:

Point 1: This reviewer understands that changing mouse facilities can affect phenotypes, however, it would be necessary to put the updated data on tumor incidence from the new facility in the revised manuscript (the graph was presented only for the reviewer).

Point 2: According to this reviewer, the lack of the HPV8/wt control group with anti-CSFR1 antibody treatment is still an important experiment to include and the reasons provided are not sufficient to exclude it. Without this group, the conclusions regarding the downstream effects described, such as decrease in keratinocyte proliferation or blood vessels, cannot be attributed to Activin expression. This point must be experimentally addressed or the conclusions should be altered accordingly.

IF for CD206 shown in Figure 6B does not represent the numbers shown in Figure 6C. Specifically, the graph shows 2-6% of CD206+ cells/area, however, the image would suggest a much higher percentage in the IgG group.

Point 4: The point regarding the expression levels of inhba was meant to be compared between skin and tumor of HPV8/wt mice not to the skin of wt/wt mice. This point should be discussed.

2nd Revision - authors' response

25 October 2016

Referee #3 (Remarks):

Unfortunately, only some comments have been appropriately addressed as detailed below.

Our response: We respectfully disagree with this comment, since we have addressed all concerns of this reviewer in our revised manuscript. In fact, most issues had been addressed by performing additional experiments. The only exception is the control experiment with anti-CSFR1 antibody and our response to this comment is detailed below.

Point 1: This reviewer understands that changing mouse facilities can affect phenotypes, however, it would be necessary to put the updated data on tumor incidence from the new facility in the revised manuscript (the graph was presented only for the reviewer).

Our response: This is an excellent suggestion and we now show an updated version of this graph (including more mice and a statistical analysis) in Fig EV3A. We also mention in the text that the change in facilities only affected the onset of tumor development in mice of both genotypes, but not the strong pro-tumorigenic effect of activin and its effect on different immune cells (page 12).

Point 2: According to this reviewer, the lack of the HPV8/wt control group with anti-CSFR1 antibody treatment is still an important experiment to include and the reasons provided are not sufficient to exclude it. Without this group, the conclusions regarding the downstream effects described, such as decrease in keratinocyte proliferation or blood vessels, cannot be attributed to Activin expression. This point must be experimentally addressed or the conclusions should be altered accordingly.

Our response: As mentioned in our reply to the initial concerns, spontaneous tumor development in HPV8/wt mice occurs very late, and in our new facility it occurs even later. Even at late time points, the percentage of HPV8/wt mice with tumors is low in the new facility (Fig EV3A, new version). Therefore, treatment of HPV8/wt mice at the same age as HPV8/Act mice and for the same period of time would not allow us to observe any effect on tumor formation. In addition, any effect of infiltrated/accumulated macrophages on other parameters could not be studied, since there is no macrophage accumulation in the skin of HPV8/Act at this early age. In order to study the effect of the anti-CSFR1 antibody on tumor formation in HPV8/wt mice we would have to inject the mice for several months and the mice would suffer too much from two injections per week during such a long period. We would not get permission from the veterinary authorities to do such an experiment. In particular, macrophage depletion over several weeks/months is likely to cause various systemic alterations, which may affect the interpretation of the data. We now mention this problem in the text. In particular, and as requested by the reviewer, we mention this limitation of our experiment in the Results (page 19, first paragraph) and we formulate our conclusion more carefully. However, we also point out that the *HPV8* transgene alone neither affected the gene expression pattern in macrophages nor angiogenesis prior to the appearance of skin tumors. Therefore, these effects are attributed to activin and not to the *HPV8* transgene.

IF for CD206 shown in Figure 6B does not represent the numbers shown in Figure 6C. Specifically, the graph shows 2-6% of CD206+ cells/area, however, the image would suggest a much higher percentage in the IgG group.

Our response: This is an important point and we therefore re-checked the analysis. We initially normalized the value to the total area of ear in this figure (including cartilage, epidermis, hair follicles). By contrast, we had normalized it to the total area of dermis in the CD68 analysis. This has now been done for CD206 and the value is now clearly higher and consistent with the values obtained for CD68. We also specify in the legend to the figure that the CD68- and CD206-positive areas were normalized to the area of dermis. We apologize for this confusion and we thank the reviewer for having detected this discrepancy.

*Point 4: The point regarding the expression levels of *inhba* was meant to be compared between skin and tumor of HPV8/wt mice not to the skin of wt/wt mice. This point should be discussed.*

Our response: We also compared the *Inhba* mRNA levels between skin and tumor of HPV8/wt mice. Interestingly, there was no significant upregulation of *Inhba* in the skin of HPV8/wt mice compared to control mice prior to tumor formation, which correlates with a lack of macrophage infiltration in the pre-tumorigenic skin (see Fig. 3). However, *Inhba* mRNA levels are increased in the established tumors (papillomas) of HPV8/wt mice compared to pre-tumorigenic (and already hyperplastic) skin of these mice. Therefore, it seems likely that upregulation of endogenous activin occurs only when the tumors appear in this tumor model. We now clarify this further in the text (page 6, second paragraph).

Editorial requests:

1) Please provide a higher quality version of Figure 5J.

Our response: We replaced this figure by a higher quality version.

2) As per our Author Guidelines, the description of all reported data that includes statistical testing must state the name of the statistical test used to generate error bars and P values, the number (n) of independent experiments underlying each data point (not replicate measures of one sample), and the

actual P value for each test (not merely 'significant' or ' $P < 0.05$ ').

Our response: The number of independent experiments had already been included in the legends of the previous version (number of mice and number of samples). In most cases we had also provided the information about the statistical test and this has now been done for all figures. Finally, we added the actual P values in all figures (for all statistically significant differences). The only exception are P values below 0.0001, for which no exact P value is provided by GraphPad Prism. All these values are now indicated as $P < 0.0001$ in the figures. This was done in the same way in other EMBO Mol Med publications (see for example Salta et al., EMBO Mol Med 8, No.9, 2016). For the calculation of the exact P values we had to recalculate the P values using a new version of GraphPad Prism (version 7). At the same time we verified all the graphs looking at the original data and the newly calculated P values. Thereby we noticed a mistake in Fig EV2G. When we initially wrote the manuscript, we showed data from epidermis and dermis in Fig 2, but then noticed that the figure is getting too crowded. Therefore, we moved the dermis data to Fig EV2. Unfortunately, at this step, we had moved the Treg epidermis data to Fig 2G instead of the Treg data for the dermis. We noticed this upon re-checking all graphs and comparison of the text and figure legends with the graphs. The correct graph has now been inserted. We strongly apologize for this mistake – we can of course send all original data upon request. The text in the Results and in the Legend was correct and the revised version of course includes the correct graph in Fig EV2G.

3) We are now encouraging the publication of source data, particularly for electrophoretic gels and blots, with the aim of making primary data more accessible and transparent to the reader. Would you be willing to provide a PDF file per figure that contains the original, uncropped and unprocessed scans of all or at least the key gels used in the manuscript? The PDF files should be labeled with the appropriate figure/panel number, and should have molecular weight markers; further annotation may be useful but is not essential. The PDF files will be published online with the article as supplementary "Source Data" files. If you have any questions regarding this just contact me.

Our response: We are of course happy to submit the complete blots, and we submit a supplementary "Source Data" file, which include the original Western blots and the zymogram.

Corresponding Author Name: Sabine Werner

Manuscript Number: EMM-2016-06493